EMBO
*reports*

# Transfer of extracellular vesicle-microRNA controls germinal center reaction and antibody production

Lola Fernández-Messina[1,2,3] (iD), Ana Rodríguez-Galán[1,2], Virginia G de Yébenes[4] (iD), Cristina Gutiérrez-Vázquez[2,†], Sandra Tenreiro[5], Miguel C Seabra[5], Almudena R Ramiro[4] (iD) & Francisco Sánchez-Madrid[1,2,3,*] (iD)

## Abstract

Intercellular communication orchestrates effective immune responses against disease-causing agents. Extracellular vesicles (EVs) are potent mediators of cell–cell communication. EVs carry bioactive molecules, including microRNAs, which modulate gene expression and function in the recipient cell. Here, we show that formation of cognate primary T-B lymphocyte immune contacts promotes transfer of a very restricted set of T-cell EV-microRNAs (mmu-miR20-a-5p, mmu-miR-25-3p, and mmu-miR-155-3p) to the B cell. Transferred EV-microRNAs target key genes that control B-cell function, including pro-apoptotic BIM and the cell cycle regulator PTEN. EV-microRNAs transferred during T-B cognate interactions also promote survival, proliferation, and antibody class switching. Using mouse chimeras with Rab27KO EV-deficient T cells, we demonstrate that the transfer of small EVs is required for germinal center reaction and antibody production *in vivo*, revealing a mechanism that controls B-cell responses via the transfer of EV-microRNAs of T-cell origin. These findings also provide mechanistic insight into the Griscelli syndrome, associated with a mutation in the Rab27a gene, and might explain antibody defects observed in this pathogenesis and other immune-related and inflammatory disorders.

**Keywords** antibody production; exosomes; extracellular vesicles; germinal center (GC) reaction; microRNAs

**Subject Categories** Immunology; Membranes & Trafficking; RNA Biology

See also: **J Pérez-Boza & DM Pegtel** (April 2020)

## Introduction

Antigen-driven immune contacts between B lymphocytes and T lymphocytes are required to trigger the germinal center reaction (GC) that gives rise to high-affinity antibodies. This process plays a pivotal role in the development of humoral immunity, and its deficiency leads to immune-related pathologies. Among primary immune deficiencies, antibody production-related diseases are the most frequent and clinically relevant [1]. Communication between immune cells coordinates an effective immune response to eliminate pathogens while maintaining physiological homeostasis. Information travels through cell contact-dependent and cell contact-independent mechanisms. These mechanisms must ensure the transfer of information, usually biomolecules, between neighboring and distant cells. A classical example of a contact-dependent mechanism is the formation of immune synapses (IS), while contact-independent mechanisms include the release of cytokines and extracellular vesicles (EVs). In recent years, EVs have emerged as key modulators of cell signaling. EVs include vesicles of endosomal origin called exosomes. EVs carry lipids, proteins, and nucleic acids, in particular small RNA species such as microRNAs (miRNAs), which regulate gene expression in recipient cells [2].

The IS is a highly specialized platform that enables contact-dependent information exchange between a T lymphocyte and an antigen-presenting cell (APC). Our previous work showed that IS formation promotes the polarized transfer of EVs from the T cell to the APC. T cell-origin EVs are enriched in the T-cell receptor (TCR) [3] and functional miRNAs that down-modulate specific target mRNAs in the recipient APC [4]. Although most studies of the IS have focused on T-cell activation and function, several reports indicate that IS formation induces changes in the APC that appear to be important for regulating the immune response. Specifically, dendritic cell apoptosis is decreased after IS formation through a mechanism that depends on NF-κB and FOXO1 [5]. Recently, we demonstrated that post-synaptic dendritic cells acquire antiviral features after receiving mitochondrial DNA from T cell-derived vesicles through the IS [6].

1 Immunology Service, Hospital de la Princesa, Instituto Investigación Sanitaria Princesa, Universidad Autónoma de Madrid, Madrid, Spain
2 Intercellular Communication in the Inflammatory Response. Vascular Pathophysiology Area, Centro Nacional de Investigaciones Cardiovasculares (CNIC), Madrid, Spain
3 Centro de Investigación Biomédica en Red, Enfermedades Cardiovasculares (CIBERCV), Madrid, Spain
4 B lymphocyte Biology Lab, Vascular Pathophysiology Area, Centro Nacional de Investigaciones Cardiovasculares (CNIC), Madrid, Spain
5 CEDOC, Faculdade de Ciências Médicas, Chronic Diseases Research Centre, NOVA Medical School, Universidade NOVA de Lisboa, Lisboa, Portugal
*Corresponding author. Tel: +34 915202307; E-mail: fsmadrid@salud.madrid.org
†Present address: Ann Romney Center for Neurologic Diseases, Brigham and Women's Hospital, Harvard Medical School, Boston, MA, USA

In the case of B cells, interactions with CD4[+] T cells are critical to trigger the germinal center (GC) response. In the dark zone of GCs, B cells engage in intense proliferation coupled to somatic hypermutation (SHM). In the light zone, higher affinity B-cell clones generated by SHM are competitively selected by interaction with follicular T helper cells. In addition, GC B cells can exchange their primary immunoglobulin (Ig) IgM/IgD B-cell receptors by class switch recombination (CSR), giving rise to B cells with specific Ig isotypes and therefore specialized effector functions. As a result of these events, the GC produces high-affinity antibody-secreting plasma cells and long-lived memory B cells [7].

MiRNAs play an important role in the regulation of B lymphocyte development, maturation, and function, including B-cell activation, malignant transformation, the generation of GC B cells, and antibody production [8,9].

In this study, we investigated the effects of EV transfer from CD4[+] T cells to B lymphocytes during IS formation, focusing on miRNA delivery and the impact of transferred miRNAs on the fate and function of recipient B cells. Using an *in vitro* model that enables IS formation between OVA-specific OT-II CD4[+]T cells and miRNA-deficient DICER-KO B cells, we identified 3 EV-miRNAs that are shuttled from the T cell to the B cell in the context of the IS and contribute to CSR and proliferation in post-synaptic B cells. In addition, we found that T to B EV transfer is critical for GC progression and antibody secretion *in vivo*. Our results provide mechanistic insight into a novel mode of regulating B-cell adaptive immune responses based on T-cell EV-miRNA transfer during IS formation, which may be involved in immune maladies associated with abnormal antibody production. A better understanding of the mechanisms underlying GC reaction dynamics may pave the way to new therapeutic strategies to modulate humoral responses.

# Results and Discussion

## IS formation promotes class-switch, proliferation, and survival of B lymphocytes

To assess whether miRNAs are specifically transferred from the CD4[+] T cell to the B lymphocyte during IS formation and potentially impact B-cell fate and function, the following model was established. Briefly, we co-cultured CD4[+] T cells isolated from OT-II mice (which express an OVA-specific transgenic T-cell receptor) with pre-activated B cells from CD19-Cre[Ki/+] Dicer[fl/fl] mice (DICER-KO). The B cells in these mice are DICER-deficient and thus unable to produce mature miRNAs, allowing identification of exogenously transferred miRNAs after IS formation (Fig EV1A). Formation of a functional, mature IS in the presence of OVA peptide, hereafter OVA, was assessed by confocal microscopy (Fig 1A), revealing an increase in the percentage of B cells forming conjugates, TCR, and actin recruitment to the contact zone between the CD4[+] T cell and the B lymphocyte, and polarization of the T-cell microtubule-organizing center (MTOC) toward the IS (Fig 1B–E).

To analyze the effects on B-cell activation and class switch recombination (CSR) *in vitro*, we used alternative B-cell stimuli before co-culture: LPS plus IL-4 to activate Toll-like receptors or CD40 plus IgM to engage the B-cell receptor (Fig EV1A). OT-II T:B cell co-culture in the presence of OVA boosted T-cell proliferation in both setups, especially in the LPS plus IL-4 condition (Fig EV1B and C). Co-culture also affected B cells, with the number of IgG1-expressing B cells significantly increased after 24 h OVA exposure (Fig 1F). Importantly, co-culture with OT-II T cells and OVA increased the percentage of proliferating B cells, as measured by Ki67 staining and cell violet tracer dilution (Fig 1G and H) as well as B lymphocyte survival 72 h after IS (Fig 1I), correlating with a lower percentage of apoptotic B cells, expressing cleaved caspase-3 (Fig 1J). Thus, the formation of a mature IS promoted CSR to the IgG1 class and promoted B-cell survival and proliferation *in vitro*.

## IS-dependent transfer of a specific set of miRNAs from the T to the B lymphocyte

Sequencing of the small RNA content in DICER-KO B cells after T-cell co-culture identified a very restricted set of 6 miRNAs significantly increased in B cells upon IS formation in the presence of OVA (Fig 2A and B, Dataset EV1). The transfer of such a specific set of miRNAs implies either that shuttled EVs bear only these specific miRNAs or more likely, that these miRNAs, which may be important for B-cell function, are stabilized in post-synaptic B cells, while the rest are degraded and/or expelled.

We focused on the 3 miRNAs conserved in humans: mmu-miR-20a-5p, mmu-miR-25-3p, and mmu-miR-155-3p (Fig EV2A). It is worthwhile mentioning that, although we focused on the upregulated miRNAs identified in our study, it is likely that other T-cell EV-miRNAs may also have a role in recipient B lymphocytes. Although DICER-KO B cells can retain low levels of miRNAs, as previously described [10,11], small RNA sequencing revealed that the increased levels of mmu-miR-20a-5p, mmu-miR-25-3p, and mmu-miR-155-3p were only found after IS formation (Fig EV2C). In addition, we observed no changes in residual Dicer expression in DICER-KO B cells upon OVA exposure (Fig EV2B). Moreover, in the absence of T-cell contact, B cells from DICER-KO (CD19Cre[Ki/+] Dicer[fl/fl]) mice showed very low levels of mmu-miR-20a-5p, mmu-miR-25-3p, and mmu-miR-155-3p (Fig EV2D) both before and after stimulation with LPS and IL-4. Also, mmu-miR-20a-5p and mmu-miR-155-3p increased upon LPS+IL-4 B-cell stimulation in WT B lymphocytes but not in DICER-KO B cells, while mmu-miR-25-3p was downregulated after stimulation, in agreement with previously published data [12]. Thus, miRNA upregulation in DICER-deficient B cells cannot be attributed to B-cell stimulation, and instead is triggered by the OVA-specific IS.

Importantly, these miRNAs are expressed in activated T cells from wild-type C57BL/6 mice and in their secreted small EVs (Fig 2C and D, Dataset EV2). Mmu-miR-20a-5p, mmu-miR-25-3p, and mmu-miR-155-3p could also be detected in small EVs derived from OT-II CD4[+] T-cell cultures, isolated from CD63- and CD81-expressing size-exclusion chromatography fractions and ultracentrifuged supernatants but not in culture medium alone, excluding possible serum contaminations in these samples (Fig EV3). The bias of small RNA sequencing protocols for particular miRNAs should be taking into account when analyzing the EV-enrichment scores (Fig EV2A).

Quantitative real-time PCR (qRT–PCR) of mmu-miR-20a-5p, mmu-miR-25-3p, and mmu-miR-155-3p confirmed increased content after IS formation in DICER-KO B-cells pre-activated with LPS plus IL-4 (Fig 2E) and especially after pre-activation with CD40 plus IgM (Fig 2F). Mmu-miR-20a-5p and mmu-miR-25-3p were significantly

more abundant in EVs than in their secreting cells (Fig EV3A), in agreement with the existence of specific mechanisms for miRNA sorting into EVs [13,14]. Accordingly, the 3 identified miRNAs are upregulated in CD4$^+$ activated T cells and effector T-cell subsets and are expressed in follicular helper T cells [12,15]. A recent report indicates that abundance of these miRNAs also increases during differentiation to antibody-producing plasma B lymphocytes in humans [16]. Activated B lymphocytes also secrete miRNA-containing EVs. However, given the low levels of mature miRNAs in DICER-KO B cells (Fig EV2) and previous work demonstrating the

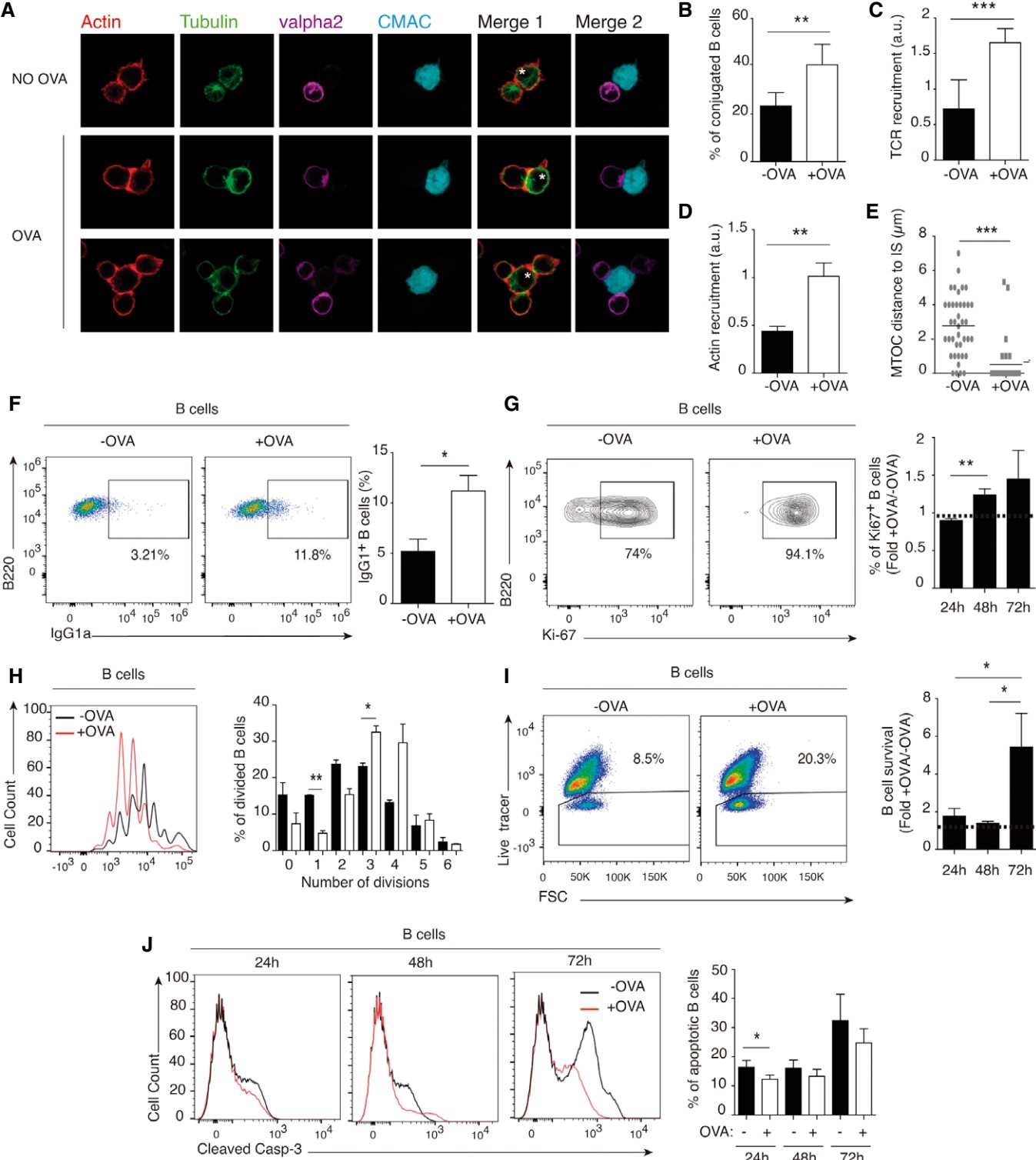

Figure 1.

◀

**Figure 1.  Mature IS formation between DICER-KO B cells and OT-II T cells promotes class switching, proliferation, and survival.**

DICER-KO B cells were co-cultured for 24 h with OTII-derived T cells in the presence or absence of OVA.

A  Representative confocal microscopy images of CMAC-stained conjugates formed between B lymphocytes (isolated from DICER-KO mouse splenocytes and pre-activated with a mixture of LPS plus IL-4) and CD4$^+$ T cells (from OVA-specific OT-II transgenic mice); conjugates were formed either in the presence or in the absence of OVA. A representative experiment of at least three independent experiments is shown.

B–E  Quantification of three representative experiments as in (A), showing the percentage of B cells forming conjugates (B), the accumulation at the IS of the TCR (C) and actin (D) at the IS, and the distance from the CD4$^+$ T-cell MTOC to the contact site with the B cell (E). Conjugates were assessed using the ImageJ plug-in for IS analysis, and MTOC–B-cell distance was measured with Imaris analysis software. Bar charts are representative of the mean of $n \geq 7$ cells analyzed $\pm$ SEM from at least 3 independent experiments. Significance was assessed by unpaired Student's $t$-test; *$P < 0.05$, **$P < 0.01$, ***$P < 0.001$.

F–J  Flow cytometry analysis of B lymphocytes co-cultured with OT-II isolated T cells in the presence or absence of OVA, showing (F) IgG1 expression after 24 h of co-culture, (G and H) B-cell proliferation after 24, 48 and 72 h co-culture (Ki67 expression in live B cells and cell violet tracer dilution, respectively), (I) survival (staining with the live marker DAPI), and (J) apoptosis (cleaved Caspase-3 expression) 24, 48 and 72 h after IS formation.

Data information: Dot plots are representative of $\geq 4$ independent experiments 48 h after IS formation, and bar charts show mean values $\pm$ SEM of at least three independent experiments at the indicated time points. Significance was assessed by paired Student's $t$-test; *$P < 0.05$, **$P < 0.01$, ***$P < 0.001$.

unidirectionality of IS-dependent EV transfer [4], we have focused our study on EVs released by T lymphocytes.

*In silico* target analyses for these miRNAs identified putative mRNA targets with pro-apoptotic effects, for example, BCL2L11 (BIM). The prediction algorithms also identified molecules that participate in B-cell homeostasis downstream of BCR signaling, for example, Pten, and several cell cycle regulators, including Tp53 and CCND1 and cyclin-dependent kinases, with important roles in GC reaction, such as CDKN1C/p57 [17] (Appendix Table S1).

qRT–PCR experiments revealed that some of these putative target mRNAs were downregulated upon IS formation in the presence of OVA (Figs 2G and H, and EV2E and F). In particular, increased miRNA transmission correlated with downregulation of molecules crucial for B lymphocyte biology, such as BIM and PTEN, which decreased more steeply in the CD40 plus IgM (Fig 2H) than in the LPS plus IL-4 B-cell co-cultures (Fig 2G). However, other predicted targets did not change their expression levels, for example, TP53 and MDM2 (Fig EV2E and F). Notably, the down-modulated targets of these miRNAs are involved in B-cell proliferation, survival, and GC reaction [17].

To characterize the function of transferred miRNAs in recipient B cells, we nucleofected B cells from wild-type C57BL/6 mice with synthetic mimic miRNAs fluorescently labeled with FAM. qPCR analysis confirmed the specific increase in mature miRNAs in FAM$^+$ FACS-sorted nucleofected cells (Fig 2I). In addition, qRT–PCR detected specific downregulation of BIM mRNA after nucleofection with mmu-miR-20a-5p or mmu-miR-25-3p but not mmu-miR-155-3p (Fig 2J), in accordance with algorithm-based predicted targets (Appendix Table S1). Likewise, PTEN mRNA levels decreased with all 3 transferred miRNAs, as predicted. FAM$^+$ B cells nucleofected with mmu-miR-20a-5p had increased switched IgG1 expression (Fig 2K), increased proliferation rate detected by cell violet tracer dilution and Ki67 expression (Fig EV4A–C), and survival (Fig EV4D). Interestingly, miR-20a-5p has been found to be upregulated in the early phases of B-cell differentiation from memory B cell to pre-plasmablast, subsequently decreasing during the transition from plasmablast to plasma cells [16]. The transfer of this specific miRNAs via the IS may thus contribute to plasma cell differentiation during early stages, shortly after antigen recognition.

**Figure 2.  mmu-miR-20a-5p, mmu-miR-25-3p, and mmu-miR-155-3p are transferred during IS and correlate with BIM and PTEN down-modulation in B lymphocytes.**                                                                                                                      ▶

CD4$^+$ T and DICER-KO B cells were co-cultured for 24 h with or without OVA, and flow cytometry-sorted B cells were analyzed by small RNA next-generation sequencing (NGS).

A  Heat map showing miRNAs significantly upregulated in B cells after co-culture with T cells in the presence of OVA. Data are from three independent experiments (samples 1–3), and only miRNAs with an adjusted $P < 0.02$ are shown.

B  Fold increase in the B-cell content of upregulated miRNAs in the presence of OVA; *$P < 0.05$, **$P < 0.01$, ***$P < 0.001$.

C  Heat map showing NGS analysis of miRNAs differentially expressed in activated WT C57BL/6 CD4$^+$ T cells and their secreted exosomes. Data are from three independent experiments (samples a–c), and only miRNAs with an adjusted $P < 0.02$ are shown.

D  Mean tags per million (TPMs) for the differentially expressed miRNAs under study in exosomes from activated CD4$^+$ T cells. Bars represent the mean $\pm$ SEM.

E, F  Quantitative real-time PCR of upregulated miRNAs after co-culture of OT-II derived CD4$^+$ T cells with DICER-KO B cells activated either with LPS+IL-4 (E) or with CD40+IgM (F). Bars represent the mean $\pm$ SEM of at least three independent experiments, and data were obtained by the $2^{-\Delta\Delta Ct}$ method using Biogazelle software. Results are expressed as a proportion of the miRNA expression in non-OVA containing co-cultures, with normalization to the small nucleolar RNAs RNU1A1 and RNU5G.

G, H  Quantitative real-time PCR showing the expression of down-modulated mRNA targets upon IS formation between OT-II CD4$^+$ T cells and DICER-KO B cells pre-activated with either LPS+IL-4 (G) or CD40+IgM (H). Results are the means $\pm$ SEM of at least four independent experiments and are expressed as a proportion of mRNA expression in the non-OVA condition, with normalization to GAPDH.

I, J  Fluorescently tagged mimics of the miRNAs mmu-miR-20a-5p, mmu-miR-25-3p, and mmu-miR-155-3p were nucleofected into activated B lymphocytes from C57/BL6 mice. Quantitative real-time PCR showing expression of nucleofected miRNA mimics in FACS-sorted FAM$^+$ B cells with normalization to RNU1A1 and RNU5G (I); and Bcl2l11 and Pten mRNA targets in mimic-nucleofected B cells with normalization to GAPDH (J). A representative experiment is shown of 4 performed, and bar charts summarize the mean $\pm$ SEM. Significance was assessed by paired Student's $t$-test comparing the plus and minus OVA conditions; *$P < 0.05$, **$P < 0.01$, ***$P < 0.001$.

K  Flow cytometry analysis of mimic-nucleofected B lymphocytes. Dot-plots show B220$^+$ IgG1$^+$ B cells 48 h after mimic nucleofection from at least four independent experiments, and bar charts show mean values $\pm$ SEM of four independent experiments.

Data information: Significance was assessed by the Student's $t$-test comparing the OVA and NO OVA conditions; *$P < 0.05$, **$P < 0.01$, ***$P < 0.001$.

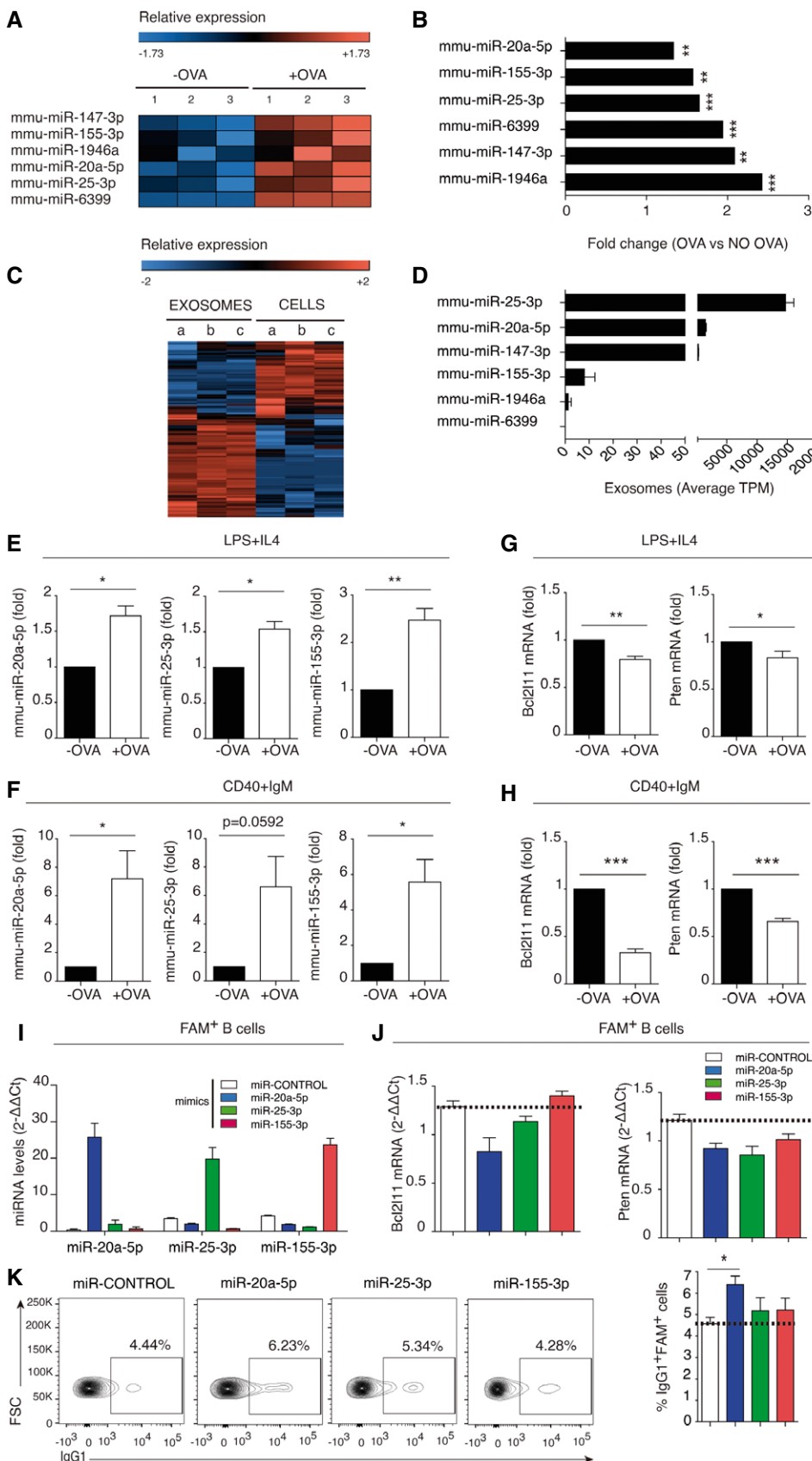

Figure 2.

Similar results were obtained using a retroviral system to express the precursor sequences of mmu-miR-25 in LPS plus IL-4-activated B cells, including increased CSR to IgG1 and increased proliferation and survival (Fig EV4E–J).

## EV transfer of CD4$^+$ T-cell miRNAs to B cells is required for efficient B-cell activation and germinal center reaction and antibody production *in vivo*

To examine the role of EV transfer in CSR to IgG1 in B cells, we blocked exosome release from CD4$^+$ T cells before IS formation using both genetic and chemical approaches. First, OTII-derived cells were nucleofected with a small-interfering RNA targeting Rab27a, a critical mediator of the exosome secretory pathway [18–21]. After IS formation using siRab27a-treated T cells, miRNA upregulation was reduced in co-cultured DICER-KO B cells (Fig 3A), correlating with a slight increase in *Bcl2l1* and *Pten* mRNAs (Fig 3B) and a reduced proportion of IgG1$^+$ cells (Fig 3C). B-cell activation was similarly inhibited after IS formation with T cells treated with manumycin A, a chemical inhibitor of neutral sphingomyelinase 2 that suppresses exosome biogenesis and secretion [4,6] (Fig EV5A–C). We conclude that EV-miRNA transfer impacts activation and differentiation of primary B cells *in vitro*.

To evaluate the role of T-cell EVs in establishing the GC reaction *in vivo*, we conducted adoptive transfer experiments. CD4$^+$ T cells were isolated from wild-type C57BL/6 mice and from mice deficient for Rab27a and Rab27b (Rab27-KO), and B cells were obtained from WT C57BL/6 mice. Isolated T and B cells were co-transferred into sublethally irradiated Rag1-KO mice, which lack both B and T mature lymphocytes. Reconstituted mice were inoculated with sheep red blood cells, and serum and spleens were analyzed after 7 days (Fig 3D). Repopulation of the B- and T-cell compartments was indistinguishable in mice adoptively transferred with Rab27-KO or WT T cells (Fig 3E). Assessment of B-cell function revealed that GC Fas$^+$ GL7$^+$ B cells were less abundant in mice reconstituted with Rab27-KO-derived T cells than in those reconstituted with WT T cells, indicating that T cell-derived EV transfer contributes to GC generation and maintenance *in vivo* (Fig 3F). Furthermore, GC Fas$^+$GL7$^+$ B cells of mice adoptively transferred with Rab27-KO CD4$^+$ T cell had an altered dark zone: light-zone ratio (Fig 3G), consistent with dysfunctional GC dynamics. Finally, serum ELISA showed a significant reduction in the production of both soluble IgM (Fig 3H) and class-switched IgG (Fig 3I), revealing the importance of EV-dependent B- and T-cell communication for antibody responses *in vivo*. Interestingly, a human immunodeficiency featuring decreased antibody production described by Griscelli *et al* [22] is associated with a mutation that causes deficiency in Rab27a. This defect is not B cell-intrinsic, but instead secondary to impairment of CD4$^+$ T-cell helper function. In this regard, our results suggest that the antibody defects in Griscelli syndrome are likely the result of deficient exosome delivery from T cells during their contact with B cells. T-cell Rab27a deficiency also underlies the Ashen mouse model, in which cytotoxic T lymphocytes display diminished lytic functions due to a Rab27a splicing mutation that causes defective granule exocytosis [23,24]. Moreover, Rab27KO mice have deficient neutrophil chemotaxis and recruitment [20], impaired granule exocytosis in neutrophils [21] and mast cells [25], and a chronic inflammatory phenotype [26]. The deficiency in the B-cell compartment identified in the present study warrants further investigation.

We report a novel mechanism for the regulation of the B-cell response through the directional transfer of a surprisingly very restricted set of T-cell exosomal miRNAs (mmu-miR-20a-5p, mmu-miR-25-3p, and mmu-miR-155-3p) during B:T cell IS. These exosomal miRNAs modulate key mRNA targets in B cells (including BIM and PTEN) and promote GC regulation, including CSR, B-cell proliferation, and survival (Fig 3J). In accordance with our data, the combined downregulation of BIM and PTEN has been shown to promote the GC reaction both in double heterozygous *hCD2-iCre*; *Pten*$^{fl/+}$;*Bim*$^{+/-}$ mice and in transgenic mice over-expressing the miR-17~92 cluster that targets both proteins [17]. Also, a higher proliferation and reduced activation-induced cell death were observed, revealing the key role that modulation of these molecules plays in T cell-dependent immune responses. In agreement with our findings, these miRNAs are 2- to 4-fold more abundant in GC B cells than in mature B cells [12]. Interestingly, the B lymphotropic Epstein–Barr (EBV) virus has been shown to promote infected naïve B lymphocyte migration to the GC [27]. This correlates with high expression of virally encoded miRNAs that target several mRNAs, including PTEN, which undergoes dramatic decreases at the protein level upon infection [28].

**Figure 3. EV transfer is required for full B-cell activation and antibody production *in vivo*.**

A  Quantitative real-time PCR (RNU1A1 and RNU5G-normalized) showing blockade of miRNA transfer from CD4$^+$ T cells pre-treated with siRNA targeting the exosome secretion mediator Rab27. Bar charts represent the mean values ± SEM of at least four independent experiments.

B  Quantitative real-time PCR showing mRNA target down-modulation after IS formation between DICER-KO B cells and CD4$^+$ T cells nucleofected with either siRab27 or scrambled control siRNA, with normalization to GAPDH. Bar charts represent the mean values ± SEM of at least four independent experiments.

C  Representative dot-plots, showing IgG1 expression of gated B220+ B cells after IS formation as in (G); the bar chart shows the mean fluorescence Intensity ± SEM and quantification of three independent experiments is shown.

D  Experimental design. Rag1KO mice were adoptively transferred with WT B lymphocytes and either Rab27-KO T cells (top) or WT T cells. Thereafter mice were inoculated with sheep red blood cells, and spleens and serum were harvested after 7 days.

E  Representative plot (left) and quantification (right) of the reconstitution of the B-cell and T-cell compartments after adoptive transfer. The graph shows the mean percentage of CD19- and CD4-positive cells from $n \geq 5$ reconstituted mice ± SEM.

F  Representative flow cytometry plot (left) and quantification (right) of GL7 and CD95/Fas expression in B cells. The graph shows the mean percentage of GL7$^+$Fas$^+$ cells from $n \geq 5$ reconstituted mice ± SEM.

G  Representative dot-plot analysis showing CXCR4 and CD86 expression on B cells for dark zone (DZ) and light zone (LZ) analysis (gated on GC GL7$^+$ Fas$^+$ B cells). The graph shows the mean percentage of DZ (CXCR4$^+$CD86$^-$) and LZ (CXCR4$^-$CD86$^+$) populations from $n \geq 5$ reconstituted mice ± SEM.

H, I  ELISA quantification of IgM (H) and IgG (I) in serum from immunized adoptively transferred mice. The graph shows the mean serum concentration of IgM and IgG from $n \geq 5$ reconstituted mice, respectively ± SEM.

J  Proposed model of exosomal miRNA transfer during cognate B- and T-cell interactions.

Data information: Significance was assessed by unpaired Student's *t* test comparing the OVA and NO OVA conditions; *$P < 0.05$.

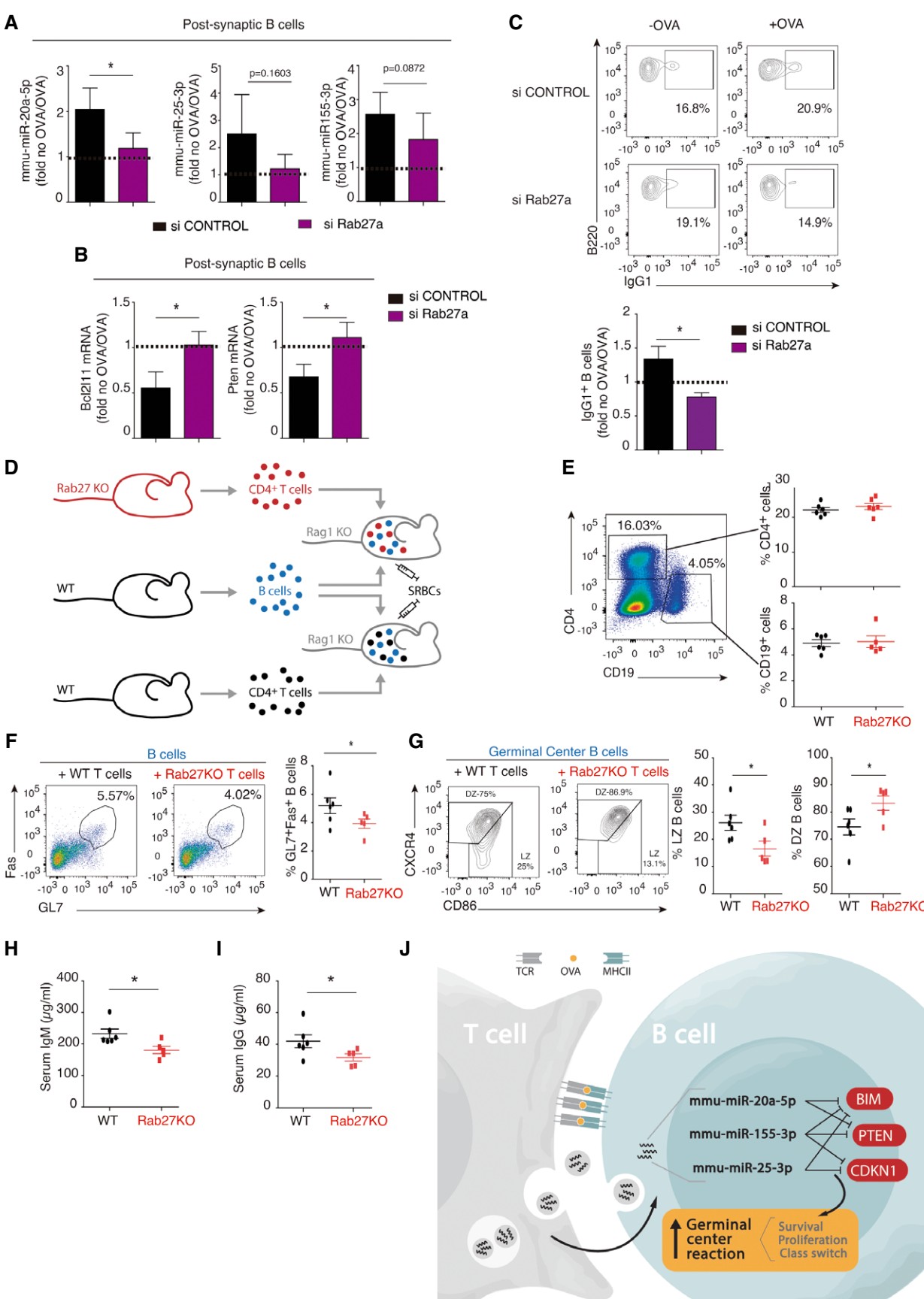

Figure 3.

Our study thus represents an additional layer of complexity in the regulation of immune responses during IS formation, and a better understanding of the mechanisms underlying this process may allow manipulation of early IS events for the treatment of immune diseases.

# Materials and Methods

### Animals

B cell-specific DICER-deficient (CD19-Cre[ki/+]Dicer[fl/fl]) mice (DICER-KO mice) were generated by crossing Dicer[fl/fl] mice with CD19-Cre[ki/+] [10,11,29] mice. DICER-KO, WT C57BL/6m, and OT-II mice were housed in specific pathogen-free conditions according to European Commission recommendations at the Centro Nacional de Investigaciones Cardiovasculares (CNIC) animal facility in Madrid. Experiments were performed with male and female mice aged 7–14 weeks. Rab27a/b double KO mice [19] and WT littermates used as a CD4[+] T-cell source in adoptive transfer experiments were generated by Miguel C. Seabra and housed in specific pathogen-free conditions according to European Commission recommendations at the Champalimaud Centre for the Unknown in Lisbon. All experimental methods and protocols were approved by the CNIC and the Comunidad Autónoma de Madrid and conformed to European Commission guidelines and regulations.

### *In vitro* cell culture, antibodies, and reagents

Primary cells were isolated by mechanical disruption of cell suspensions of spleen (and also lymph nodes in the case of CD4[+] T-cell isolation). B cells from DICER-KO and C57/BL6 wild-type mice were isolated by immunomagnetic depletion with anti-CD43 beads (Miltenyi). During the 24 h before co-culture, isolated B cells were cultured in EV-depleted complete RPMI medium [EV-free RPMI supplemented with 10% EV-depleted fetal bovine serum (discarding the pellet after ultracentrifugation for 16 h at 100,000 *g*), 50 µM β-mercaptoethanol (Invitrogen), 10 mM Hepes (Invitrogen), and antibiotics]. The same batch of ultracentrifuged EV-depleted serum was used for all experiments to minimize variability and checked by NanoSight for EV depletion. For B-cell pre-activation, this medium was supplemented either with 25 µg/ml LPS (Sigma) and 10 ng/ml IL-4 (Peprotech) or with 10 µg/ml CD40 and 10 µg/ml F(ab)'2 fragments of goat anti-mouse IgM (Jackson Immunoresearch). Regardless of the activation method, the B cells were incubated in the presence or absence of 5 µg/ml OVA peptide 323–339 (GenScript). For OT-II-derived CD4[+] T-cell isolation, splenocytes and lymph node cells were incubated for 16 h in EXO-free RPMI medium supplemented with 5 µg/ml of OVA, and T cells were then purified using the CD4[+] T-cell isolation kit (Miltenyi Biotec). Antibodies and primers are listed in Appendix Table S2.

### Immune synapse co-cultures

Pre-activated DICER-KO B cells and CD4[+] T cells isolated from OT-II mice were co-cultured at a 1B: 4T cell ratio in EXO-free RPMI medium either in the presence or absence of 5 µg/ml OVA peptide 323–339. After the times indicated in each individual experiment, B cells were isolated by flow cytometry cell sorting.

### Confocal microscopy

B cells isolated from DICER-KO mice were pre-activated for 24 h, washed once with HBSS, and stained with 10 µM cell-tracker blue CMAC (Invitrogen). The B cells ($6.25 \times 10^4$) were washed and resuspended in 100 µl EXO-free RPMI complete medium and mixed with 250,000 OT-II-isolated CD4[+] T cells (as described above for co-cultures) on poly-L-lysine-coated coverslips and allowed to settle for 1 h at 37°C. Thereafter, conjugates were fixed with 4% paraformaldehyde for 20 min and blocked and permeabilized at room temperature with 0.2% Triton X-100 in staining buffer [60 mM PIPES, 25 mM Hepes, 5 mM EGTA, 2 mM $MgCl_2$, 3% bovine serum albumin, 100 µg/ml γ-globulin, and 0.2% azide]. Cells were then incubated in staining buffer containing 5 µg/ml Alexa Fluor 568-conjugated phalloidin, 0.1 µg/ml fluorescein isothiocyanate (FITC)-conjugated anti-α-tubulin, and biotinylated anti-vα2 antibody followed by streptavidin-labeled Alexa Fluor 647. The fixed and stained conjugates were mounted in Prolong Gold and analyzed under a Leica SP5 confocal microscope (Leica) fitted with a HCX PL APO × 63/1.40–0.6 oil objective. Images were processed and assembled using ImageJ (https://imagej.nih.gov/ij/) and Photoshop. For quantification in an individual IS, we used a specifically designed ImageJ plugin for synapse measurements, as described in [30]. MTOC polarization to the IS was assessed by measuring the distance from the CD4[+] T-cell MTOC to the contact site with the B cell using Imaris software.

### RNA extraction, real-time PCR, small RNA next-generation sequencing (NGS), and data analysis

Flow cytometry-purified cells or post-synaptic CD4[+] T cell-derived EVs were lysed in QIAZOL lysis buffer, and RNA was extracted using the miRNeasy Mini Kit (QIAGEN).

For small RNA sequencing, RNA integrity was checked using an Agilent 2100 Bioanalyzer (Agilent) for total RNA (RNA nano-chips) and for small RNA (small RNA chips), and concentrations were measured in a Nanodrop-1000 and using the Quantifluor RNA system (Promega). Three independent experiments were analyzed, with an RNA integrity number (RIN) ranging from 9 to 10, and small RNA libraries were generated using the NEBNext Small RNA Library Prep Set from Illumina. Single read NGS was performed using an Illumina HiSeq 2500 System.

Small RNASeq data were analyzed by the Bioinformatics Unit at CNIC. Sequencing reads were processed with a pipeline that used FastQC, to assess read quality, and Cutadapt to trim sequencing reads, eliminating Illumina adaptor remains, and to discard reads that were shorter than 20 bp or had not been trimmed at all. Resulting reads were aligned against a mouse sequence database consisting of the mature miRNA component of miRBase21, using parameters optimized for very short sequence alignments, with either bowtie or BWA. Expression was then quantified with RSEM [31], to obtain matrices consisting in either raw expected counts or TPM (transcripts per million). Raw counts were processed with an analysis pipeline that used Bioconductor package EdgeR [32] for normalization (using TMM method, with parameter

log-ratioTrim = 0.4) and differential expression testing, taking into account only those small RNAs expressed at a minimal level of 1 CPM in a number of samples equal to the number of replicates of the condition with less replicates. When required, a blocking variable was used to define groups of samples that were expected to be similar. Changes in small RNA expression were considered significant when Benjamini and Hochberg adjusted $P < 0.2$.

RNA was retrotranscribed using either the miRCURY LNA Universal RT miRNA PCR System (EXIQON) for miRNA or the Promega RT kit for mRNA. Real-time (RT)–PCR was performed in a CFX384 Real-time System (Bio-Rad) using SYBR Green PCR Master Mix (Applied Biosystems). Reactions were analyzed with Biogazelle QbasePlus software. Mature miRNA levels were normalized to the small nucleolar RNU1A1 and RNU5G [13], whereas mRNA levels were normalized to glyceraldehyde-3-phosphate dehydrogenase (GAPDH) and expressed as a relative variation from control levels. All primers for miRNA analysis were purchased from EXIQON, and primers used for mRNA detection (Metabion) are listed in supplementary methods.

### In silico target analysis

Putative mRNA targets for human conserved miRNAs upregulated in post-synaptic B lymphocytes, summarized in Appendix Table S1, were identified using the prediction algorithms miRTarBase and miRanda.

### T-cell culture and EV recovery from supernatants

For small RNA sequencing, total splenocytes were isolated from spleens of 8- to 12-week-old WT C57BL/6 mice. After erythrocyte lysis, cells were cultured in RPMI supplemented with 10% FBS, 50 μM β-mercaptoethanol, 1 mM sodium pyruvate, and 2 μg/ml concanavalin A. After 36 h, cells were washed and cultured in RPMI supplemented with 10% EXO-free FBS and 50 U/ml mrIL-2. Medium was refreshed every 2 days. EVs were isolated from supernatants of 6-day lymphoblast cultures by serial ultracentrifugation [6]. Briefly, cells were pelleted and the supernatant centrifuged at $2,000\times g$ for 20 min. The collected supernatant was subsequently ultracentrifuged at $10,000\times g$ for 40 min at 4°C (Beckman Coulter Optima L-100 XP, Beckman Coulter), followed by a final ultracentrifugation at $100,000\times g$ for 1 h at 4°C. EVs were resuspended in PBS for NanoSight analysis or in Laemmli loading buffer for Western blot analysis [33]. RNA was isolated as described above.

For EV characterization, splenocytes and lymph nodes were harvested from OT-II mice as described above. Supernatants from 72 h CD4[+] T-cell cultures were processed by ultracentrifugation as previously described, followed by PBS wash, and size-exclusion chromatography, as described in [34] for analysis of small EVs by CD3 and CD81 dot-plot and miRNAs content by qPCR, as described above. For these samples, as non-EVs do not express or express very low levels of small nucleolar RNAs, the spike-in UniSp6 was used for normalization.

### Flow cytometry and cell sorting

B cells were incubated with the cell tracer CMAC before IS co-culture. Alternatively, B cells were stained after IS co-culture with APC-conjugated B220 antibody (BD Pharmingen). B cells were then isolated by flow cytometry cell sorting in a FACSAria cell sorter, with elimination of dead cells by DAPI or propidium iodide staining. Retrovirus-transduced B cells were isolated for RNA extraction and analysis by sorting for GFP[+] cells.

### Exosome inhibition

To inhibit exosome secretion, CD4[+] T cells isolated from OT-II mice were pre-incubated with 2 μM manumycin A (Sigma) for 2 h before B-cell co-culture. Alternatively, isolated T cells were nucleofected with 250 nM of the ON-TARGET plus mouse Rab27a siRNA-SMART pool (Dharmacon), using the mouse CD4[+] T-cell nucleofector kit (Amaxa). Nucleofected cells were then resuspended in complete EXO-free RPMI medium and incubated for 24 h before IS co-culture.

### miRNA retroviral transduction

B cells isolated and activated *in vitro* in the presence of LPS plus IL-4 were transduced as described in [35]. Briefly, precursor miR-25-GFP construct vectors were cloned in pMXPIE-Btk by PCR amplification of genomic DNA isolated from mouse B cells using the Gentra Puregene cell kit (Qiagen) with specific primers to amplify the precursor mmu-miR-25 sequence and its flanking 50 bp genomic sequences obtained from miRBase. Retroviral supernatants were produced by transiently co-transfecting 293T-HEK cells with pCL-Eco (Imgenex) and pre-miRNA-GFP retroviral vectors or pMXPIE-Btk retroviral vector, using calcium phosphate or Lipofectamine 2000 (Thermo Fisher scientific). Mouse primary B cells were then transduced with retroviral supernatants for 20 h in the presence of 8 μg/ml polybrene (Sigma), 10 mM Hepes (Invitrogen), and 50 μM β-mercaptoethanol (Invitrogen). GFP[+] cells were sorted at 48 h post-transduction (FACSAria, BD Biosciences) and processed for QIAZOL cell lysis, RNA extraction, and qPCR analysis. GFP[+] cells were stained with DAPI to detect nuclei, and expression of IgG1, Ki67, and annexin was determined by flow cytometry.

### In vivo adoptive transfer experiments

C57/BL6 WT B cells and CD4[+] T cells were isolated from C57/BL6 wild-type mice and their Rab27-KO littermates (see *In vitro* cell culture, antibodies, and reagents, above). B cells ($6 \times 10^6$) and CD4[+] T cells ($4 \times 10^6$) were adoptively transferred into sublethally irradiated Rag1-KO mice. At 24 h after reconstitution, mice were inoculated with $10^8$ sheep red blood cells. After 7 days, splenocytes were harvested and processed for flow cytometry analysis and sorting of Fas[+]GL7[+] GC center and Fas[−]GL7[−] B cells and CD4[+] T cells for RNA extraction and qPCR analysis. Serum from these mice was obtained by blood centrifugation for 30 min at $13,000\times g$ and used for ELISA analysis.

### IgG and IgM ELISA

ELISA kits were purchased from Bethyl Laboratories. Absorbance was measured at 450 nm, with a reference wavelength of 570 nm.

## Data availability

Sequencing data have been deposited in the Gene Expression Omnibus and are available to readers under record GSE141045; https://www.ncbi.nlm.nih.gov/geo/query/acc.cgi?acc = GSE141045.

**Expanded View** for this article is available online.

## Acknowledgements

NGS experiments were performed in the CNIC Genomics Unit (Centro Nacional de Investigaciones Cardiovasculares, Madrid, Spain) and analyzed by the CNIC Bioinformatics Unit. We thank Dr. Miguel Vicente-Manzanares for critical reading of the manuscript and Dr. Henar Suárez and Dr. María Yañez-Mó for their help with SEC experiments. This manuscript was funded by grants SAF 2017-82886-R (FS-M) from the Spanish Ministry of Economy and Competitiveness; CAM (S2017/BMD-3671-INFLAMUNE-CM) from the Comunidad de Madrid (FS-M); CIBERCV (CB16/11/00272), BIOIMID PIE13/041 from the Instituto de Salud Carlos III and from the Fundación La Marató TV3 (grant 122/C/2015). The current research has received funding from "la Caixa" Foundation under the project code HR17-00016. VGY is supported by the AECC foundation. A.R.R. is supported by CNIC funding. This project was funded by the Spanish Ministerio de Ciencia, Innovacion y Universidades SAF2016-75511-R, and La Caixa Health Research Program HR17-00247 grant to A.R.R. Grants from Ramón Areces Foundation "Ciencias de la Vida y de la Salud" (XIX Concurso-2018) and from Ayuda Fundación BBVA y Equipo de Investigación Científica (BIOMEDICINA-2018) (to FSM). The CNIC is supported by the Ministerio de Ciencia, Innovacion y Universidades and the Pro-CNIC Foundation, and is a Severo Ochoa Center of Excellence (SEV-2015-0505).

## Author contributions

LFM planned and performed experiments, analyzed and interpreted data, and wrote the manuscript with input from all authors; ARG collaborated in mouse experiments, data interpretation, and writing of the manuscript; VGY contributed to the cloning of retroviral constructs and helped with research planning, data analysis, and manuscript writing; CGV generated the samples for the exosome NGS; ST and MCS generated and provided Rab27-deficient mice and samples for *in vivo* experiments; ARR helped with research design, provided reagents, and collaborated in data interpretation and manuscript writing; FSM planned and coordinated the research, discussed results, and supervised and contributed to manuscript writing.

## Conflict of interest

The authors declare that they have no conflict of interest.

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
