## [Review Process File · EMBO Reports]

Transfer of extracellular vesicle-microRNA controls germinal center reaction and antibody production

Lola Fernández-Messina, Ana Rodríguez-Galán, Virginia García de Yébenes, Cristina-Gutiérrez-Vázquez, Sandra Tenreiro, Miguel C. Seabra, Almudena R. Ramiro and Francisco Sánchez-Madrid

Review timeline:

Submission date:	23 July 2019
Editorial Decision:	20 August 2019
Revision received:	4 December 2019
Editorial Decision:	17 January 2020
Revision received:	21 January 2020
Accepted:	24 January 2020

Editor: Achim Breiling

Transaction Report:

1st Editorial Decision

20 August 2019

Thank you for the transfer of your research manuscript to EMBO reports. We have now received the full set of referee reports that are copied below.

As you will see, all referees think that the study is of interest, but also indicate that it remains rather preliminary, and that part of the conclusions are not supported by the data. All referees seem to agree that extensive further experimentation, in particular several control experiments, are needed to strengthen the major conclusions.

Given these comments, and considering the amount of work required to address them, we cannot offer to publish your manuscript. However, in case you feel that you can address the referee concerns in a timely and thorough manner, and can obtain experimental data that would considerably strengthen the study as outlined in the referee reports, we would have no objection to consider a new manuscript on the same topic in the future. Please note that if you were to send a new manuscript this would be assessed again with respect to the literature and the novelty of your findings at the time of resubmission.

I am sorry to have to disappoint you this time. I nevertheless hope, that the referee comments will be helpful in your continued work in this area, and I thank you once more for your interest in our journal.

REFEREE REPORTS

Referee #1:

In this original work the authors show evidence that the formation of a mature IS promotes class switch recombination to the IgG1 class switch, B cell survival and proliferation. The authors

hypothesize that secreted miRNAs may stimulate this process and show that the small RNA content in DICER-KO B cells after T cell co-culture is increased in 6 miRNAs upon IS formation in the presence of OVA specific cognate interactions. Then they show these miRNAs target genes (BIM and PTEN) involved in the GS reaction and that the transfer from t to B cells is mediated by RAB27, a known regulator of exosomes biogenesis and secretion.

Overall the experiments are well conducted, controlled and the presented results are generally convincing. I have a few remarks that should be addressed.

a) The authors use DICER-specific B cell KO mice as recipient cells. However, it is not clear what the background of miRNAs is in these cells. It is known that DICER KO cells, as opposed to Drosha KO cells, retain certain miRNAs. The authors should show that the DICER KO B cells before IS stimulation are not enriched in the particular miRNAs of interest and mention this potential caveat of using DICER-KO in the discussion.

b) The authors focus is on T cell exosomes during the IS. However it is known that activated B cells also secrete functional exosomal-miRNAs, might these have a role as well? I feel this possibility of reciprocal interaction should at least be mentioned.

c) The authors do not mention on the transfer mechanism or show direct evidence of functional transfer via exosomes with for example reporter assays as described by Pegtel et al (PNAS 2010). I understand that this is 'off topic' for a 'report' but the authors should mention that this (arguably) remains a golden standard and that the mechanism of actual functional transfer remains to be determined. Indeed, could the authors comments as to how they think the transferred miRNAs accumulate in the recipient B cells? Are they preferentially expelled or preferentially loaded into RISC in the recipient cells?

d) The authors claim that their mechanism explains some of the features of the Griscelli syndrome, in which RAB27 deficiency leads to less Ab production. However as mentioned by the authors RAB27 may also be directly (impaired Ab secretion) or indirectly (T cell help). It is intriguing that EBV-infected B cells in humans have high expression of virally-encoded miRNAs that target BIM (Marquitz et al., *Virology* 2012), on of the putative targets identified by the authors. EBV is believed to 'push' infected cells through a GS-like reaction (Thorley-Lawson et al., *Curr Top Microbiol Immunol.* 2015).

Referee #2:

The study described in this manuscript examines transfer of miRNAs via EVs from T-cells to B-cells at the immune synapse, with the conclusion that T-cell EVs shuttle functional miRNAs to B-cells. Unfortunately, the data do not support the conclusions, the study is methodologically weak, and field guidelines are not followed. As for several previous studies in this genre, a close examination of the data suggests that serum contaminants or cellular fluctuations are a more likely explanation of the observations than selective miRNA loading and transfer via EVs.

Major points:

1) It is unclear if the mouse model actually "worked" or could "work" since so many mature miRNAs persisted after the Dicer knockout. Mature miRNAs are very stable in some cells, lasting for up to several weeks. Also, the data support an apparent increase in some miRNAs even though there is little indication that EV transfer or even transfer of contaminants from cell culture is a major source of miRNAs. A careful study of miRNA persistence in this model would be a prerequisite to interpretation of the data.

2) Use of the term "exosome" in the title and throughout the manuscript is unsupported, and it seems that the involvement of EVs is more assumed than supported. It should be EVs or small EVs, but even this is not well supported without more data. In addition to the perplexing and total lack of EV characterization, no specific evidence for exosome (vs other EV) involvement is shown. The authors are encouraged to consult with Lotvall et al, *J Extracell Vesicles*, 2014 and other recommendations

of ISEV to improve design, nomenclature and reporting. Rab27a is now thought to be involved in EV biogenesis more broadly than just inside the endosome. Similarly, manumycin A and other purported exosome inhibitors have multiple modes of action and may work as well at the plasma membrane as in the endosomal system.

3) The conclusions about specific EV loading into T-cell EVs are erroneous and not supported by the methods or the data. In fact, they are contradicted by the data. The EV separation method was ultracentrifugation, Although UC is still widely used and obviously has a continuing importance in the field, it does not efficiently purify EVs from other miRNA carriers, especially when no wash steps are performed, as here. The cells were cultured in medium supplemented with what is described as "EXO-free" serum but in reality this serum was not free of exosomes or other EVs. The serum was ultracentrifuged overnight, but this procedure is unable to remove all EVs. Additionally, no mention is made of diluting the serum appropriately before dilution or confirming removal of EVs, casting more doubt on the suitability of this serum. Please consult Driedonks et al, JEV, 2019 on the difficulties of removing EVs from serum.

Even if the serum were completely EV-depleted, most miRNA in serum is not in EVs. Centrifugation methods used in this paper will give pellets that contain both EV and non-EV miRNAs.

If one examines the TPM table for T-cells versus EVs, contamination of the EV fraction with serum miRNAs is clear, see Krichevsky's paper on serum RNA artifacts. For example, red blood cell-specific miR-451 and miR-144 show up only in the EVs. Stroma-specific miR-143 and miR-145 are only in the EVs or, for miR-143-3p, at hundreds of reads in the EVs vs none or single digits in the cells. Similarly, miR-486 is highly abundant in the EVs but not in the cells. This evidence of contamination makes it impossible to credit the claim that miRNAs are specifically loaded into EVs, since one cannot know which EV miRNAs are contaminants and which are from the cells.

4) This is related to the previous point, but I cannot help but come away with the impression that there was little basis for the selection of the chosen miRNAs. Ranking the miRNAs by differential abundance in B-cells across the conditions, there were numerous miRNAs for which a better case could be made. Some of the selected miRNAs are also indicative of fundamental problems with the analysis. To give some examples:

miR-1946a was detected in all B-cell samples, and at a respectable average of just under 100 TPM. This miRNA also underwent a relatively large fold change in co-cultured versus alone. However, it was detected in only one T-cell/"EV" pair, and there at less than 10 TPM. Also, lower in the EV sample than in the T-cells. Strangely, Figure 3D still shows this miRNA and even indicates variance when there is none (only one positive sample!).

miR-155-3p was detected at >900 TPM in the B-cells, but had negligible levels in the T-cells and was detected in only 2 of 3 T-cell EV samples.

miR-6399 has an average of 72 TPM in B-cells, but was detected at single-digit TPM (3) in only one T-cell sample and no T-cell EVs.

miR-143-3p, despite having hundreds of TPMs in the EV fraction from T-cells, went from 0 to 6 in the B-cells in one experiment, 6 to 0 in another, and 49 to 74 in the third. There was no clear RNA transfer.

miR-486-5p, despite having nearly 900 TPM in the EV fraction from T-cells, increased just as much in the "alone" B-cells as in the co-culture condition. If anything, this suggests that some serum contaminants can skew the results of the experiments, and that these contaminants have more of an effect than the actual T-cell EVs.

Many miRNAs undergo a decline in OVA+ co-culture but not without co-culture. These include miR-15a, miR-191, and miR-10b. These are not mentioned or explained.

These examples suggest that much more is happening in the B-cells under these experimental conditions than EV transfer alone, and that miRNAs can undergo an apparent "increase" even in these Dicer-negative cells. Also that EV transfer from T-cells probably does not explain much if any of the observed miRNA fluctuations.

5) Sequencing data should be deposited with a public database. While the data can be embargoed until publication, a working link should be provided for reviewers. The spreadsheets are helpful, but do not replace the need to deposit raw data.

Minor points:

While many readers will be familiar with immunology, and will in any case learn more as they read

deeper into the manuscript, it would be helpful for the broader readership to define the cells and interactions more explicitly in the abstract and introduction. Instead of "T-B" and "B-T", start with "contacts between B lymphocytes and T lymphocytes" and also keep the order consistent.

The figures require more attention. For a few examples:

Bar graphs are less informative than plotting the actual data points.

Red/Green (Fig 2 A, C) should be replaced by colors that are distinguishable by the colorblind.

Figure 2B is misleading, since fold change of 1=no change. So the axis should be at 1 not 0. Better would be to show % increase. Also, variance is not shown.

Minor edits:

Intro

"while maintaining homeostasis": what homeostasis?

bio-molecules: no hyphen needed

Results

"To analyze the effects on B cell activation and class-switch recombination (CSR)

in vitro, we used alternate B cell stimuli": I believe you mean "alternative" not "alternate"

"IS-dependent transfer of a specific set of microRNAs from the T to the B lymphocyte" section: the first sentence is repeated from the intro and should be removed or shortened.

Methods

Nanosight is "NanoSight" and Laemli is "Laemmli"

Referee #3:

Fernandez-Messina et al. investigate the requirement of T cell-derived exosomal microRNAs through the immunological synapse in the activation of B cells in vitro and in germinal center (GC) development in vivo. The authors report that antigen-dependent B cell development is controlled by this mechanism, and identify a set of six microRNAs as the underlying biological mechanism for the activation. The in vivo role of the interaction is demonstrated by exosome-deficient T cells in immunized mice undergoing the GC reaction.

There are several experimental issues that affect the interpretation of the results. While conceptually interesting, as presented, the work has preliminary character.

- 1) CD19-Cre mice were used to delete Dicer in Dicer-floxed B cells. CD19-Cre mice delete the floxed allele only in about 90% of B cells, which has the implication that a sizable fraction of cells are not deleted, which affects single cell assays as the ones shown in Fig. 1A. It is necessary to demonstrate that the B cells forming immunological synapses with the T cells are indeed Dicer-deficient and do not represent contaminating non-deleted B cells.
- 2) For the experiments with cell populations, also here the incomplete deletion issue needs to be addressed, although the results would suggest that it is unlikely that the difference is due to a preferential reactivity or outgrow of the 10% non-deleted cells. A response to this critical issue would be appropriate, and it should be addressed in writing to pre-empt any thoughts from a critical reader.
- 3) Fig. 3F: GL7 staining alone cannot be used as a measure for GC B cells. It need be used in combination with CD38 or CD95, or PNA. Only double stained cells, such as GL7+CD38-low or GL7+CD95+ or PNA+ identifies GC B cells. Since the enumeration of GC B cells is the end-point of the analysis, a rigorous analysis of GC B cells numbers and percentages by flow, and GC structure by immunohistochemistry of tissue sections, would be appropriate to convince that GC development is affected.
- 4) Fig. 3G: The staining does not identify dark zone and light zone B cells correctly. Light zone B cells are CD86+CXCR4-minus, and dark zone cells CXCR4+CD86-minus. Therefore, also the summary plots are incorrect, and no conclusions can be drawn from this experiment. For how the staining should look like, please refer to the original Victora Cell and Blood papers where this staining has first been described.
- 5) Fig. 3H: Measuring total serum IgM and IgG is not the adequate approach to measure GC output. For this, the immunization with, and specific detection of antibodies against, a model antigen would be required.

Referee #1:

In this original work the authors show evidence that the formation of a mature IS promotes class switch recombination to the IgG1 class switch, B cell survival and proliferation. The authors hypothesize that secreted miRNAs may stimulate this process and show that the small RNA content in DICER-KO B cells after T cell co-culture is increased in 6 miRNAs upon IS formation in the presence of OVA specific cognate interactions. Then they show these miRNAs target genes (BIM and PTEN) involved in the GS reaction and that the transfer from t to B cells is mediated by RAB27, a known regulator of exosomes biogenesis and secretion.

Overall the experiments are well conducted, controlled and the presented results are generally convincing. I have a few remarks that should be addressed.

We very much appreciate the review and comments and are happy to read that the reviewer finds this “original work” interesting and “the experiments are well conducted, controlled and the presented results are generally convincing”. We have addressed all his/her remarks below and have performed additional experiments and modified the manuscript accordingly.

- a) The authors use DICER-specific B cell KO mice as recipient cells. However, it is not clear what the background of miRNAs is in these cells. It is known that DICER KO cells, as opposed to Drosha KO cells, retain certain miRNAs. The authors should show that the DICER KO B cells before IS stimulation are not enriched in the particular miRNAs of interest and mention this potential caveat of using DICER-KO in the discussion.

We do agree with the reviewer that CD19-Cre^{Ki/+} DICER^{fl/fl} (DICER-KO) cells can retain low levels of miRNAs, as previously described (Belver et al, 2010; Harfe et al, 2005) and also indicated by the RNA sequencing data in this study. Although we acknowledge that this could represent a limitation of the model used, small RNA sequencing revealed that the increase of the three miRNAs was found when comparing the -OVA and +OVA conditions, while low levels of these miRNAs were detected in B cells in the absence of a mature OVA-specific IS (new Expanded View Fig. EV2C).

Moreover, we have quantified by qPCR the relative levels of the specific transferred miRNAs under study in WT and DICER-KO B cells before and after stimulation with LPS and IL4 (new Expanded View Fig. EV2D). This analysis showed that isolated DICER-KO B cells exhibited very low levels of mmu-miR-20a-5p, mmu-miR-25-3p and mmu-miR-155-3p (EV2D), which were not increased upon LPS+IL4 stimulation. Also, mmu-miR-20a-5p and mmu-miR-155-3p, but not mmu-miR-25-3p, levels increased upon B cell stimulation with LPS and IL4 only in WT B lymphocytes (mmu-miR-20a-5p, fold-increase: 2,19; mmu-miR-155-3p, fold-increase: 8,16; mmu-miR-25-3p, fold-increase: 0,83). This indicates that the upregulation of the identified miRNAs after an OVA-specific IS could not be attributed just to B cell stimulation, and it clearly does not account for miRNA increase in DICER-deficient B cells. Our LPS and IL4 induction results are in agreement with previously published data, showing that the three identified miRNAs were 2-4 fold more expressed in GC B cells compared to total mature B lymphocytes (Kuchen et al, 2010).

Additionally, we have included the data showing similar levels of DICER mRNA in our experimental conditions, excluding an eventual overgrowth of DICER non-depleted cells in OVA vs non-OVA cocultures (new Expanded View Fig. EV2B).

We have added all this additional supplementary information in the figures and modified the text (page 6, lines 7-20) and figure legends accordingly, as follows: “We focused on the 3 miRNAs conserved in humans: mmu-miR-20a-5p, mmu-miR-25-3p, and

mmu-miR-155-3p (Fig EV2A). Although DICER-KO B cells can retain low levels of miRNAs, as previously described (Belver et al, 2010; Harfe et al, 2005), small RNA sequencing revealed that the increased levels of mmu-miR-20a-5p, mmu-miR-25-3p, and mmu-miR-155-3p were only found after IS formation (Fig EV2C). In addition, we observed no changes in residual Dicer expression in DICER KO B cells upon OVA exposure (Figure EV2B). Moreover, in the absence of T cell contact, B cells from DICER KO (CD19Cre^{Ki/+} Dicer^{fl/fl}) mice showed very low levels of mmu-miR-20a-5p, mmu-miR-25-3p and mmu-miR-155-3p (Fig EV2D) both before and after stimulation with LPS and IL4. Also, mmu-miR-20a-5p and mmu-miR-155-3p increased upon LPS+IL4 B cell stimulation in WT B lymphocytes but not in DICER KO B cells, while mmu-miR-25-3p was downregulated after stimulation, in agreement with previously published data (Kuchen et al, 2010). Thus, miRNA upregulation in DICER-deficient B cells cannot be attributed to B cell stimulation, and instead is triggered by the OVA-specific IS.”

- b) The authors focus is on T cell exosomes during the IS. However it is known that activated B cells also secrete functional exosomal-miRNAs, might these have a role as well? I feel this possibility of reciprocal interaction should at least be mentioned.

Most immune cells, including activated B lymphocytes, do secrete extracellular vesicles and thus in principle, the functional contribution of B-cell derived exosomal miRNAs should be considered. However, since DICER-KO B lymphocytes used in our model exhibit very low levels of mature miRNAs, specifically of the miRNAs under study (Figure EV2) and previous work in the laboratory showed, using CD63-GFP fusion proteins, that during immune synapsis, the secretory machinery polarization and vesicle transfer occurred mainly unidirectionally from the T cell to the B cell (Mittelbrunn et al, 2011), we have focused our study in T cell-derived exosomes.

We have discussed this point in the paper, in the following paragraph (page 7, lines 7-11): “*Activated B lymphocytes also secrete miRNA-containing EVs. However, given the low levels of mature miRNAs in DICER-KO B cells (Fig EV2) and previous work demonstrating the unidirectionality of IS-dependent EV transfer (Mittelbrunn et al, 2011) we have focused our study on EVs released by T lymphocytes.*”

- c) The authors do not mention on the transfer mechanism or show direct evidence of functional transfer via exosomes with for example reporter assays as described by Pegtel et al (PNAS 2010). I understand that this is 'off topic' for a 'report' but the authors should mention that this (arguably) remains a golden standard and that the mechanism of actual functional transfer remains to be determined. Indeed, could the authors comments as to how they think the transferred miRNAs accumulate in the recipient B cells? Are they preferentially expelled or preferentially loaded into RISC in the recipient cells?

Although reporter assays have been widely used to show direct evidence of functional transfer, this approach would be technically very challenging in our model with primary B cells due to the limited B cell survival *in vitro*. Nevertheless, previous work in the laboratory using the T lymphocyte cell line Jurkat J77 and the B cell line Raji in the presence of SEE superantigen, showed that synaptically transferred miR-335, was functional in recipient Raji B cells, and downregulated its target as assessed by reporter assays (Mittelbrunn et al, 2011). This work offers a “proof-of-concept” that exosomal miRNAs transferred from the T cell to the B cell during synapsis are functional. In our model, we used a gain-of function approach to determine the involvement of these miRNAs in mRNA target downmodulation, in combination with a strategy of T cell-exosome blockade to show that target downregulation was dependent on exosomal miRNAs.

Regarding the mechanism of “miRNA retention” in recipient B cells, this remains still unsolved, as further experimentation will be required to properly address this point.

However, since T cell derived exosomes do contain many more miRNAs than the ones identified to be upregulated upon IS formation it seems plausible to speculate “*either that transferred exosomes bear only these specific shuttled miRNAs or more likely that these miRNAs, which may be important for B cell function, are stabilized in post-synaptic B cells, while the rest are degraded*”, as discussed in the manuscript. Our data would also be compatible with a differential loading into the miRISC complex, although further experimentation would be required to dissect the mechanisms underlying this process.

We speculate that transferred miRNAs may be preferentially stabilized in post-synaptic B cells while others would be “*preferentially degraded and/or expelled*” as indicated now in the manuscript (page 6, lines 4-6).

- d) The authors claim that their mechanism explains some of the features of the Griscelli syndrome, in which RAB27 deficiency leads to less Ab production. However as mentioned by the authors RAB27 may also be directly (impaired Ab secretion) or indirectly (T cell help). It is intriguing that EBV-infected B cells in humans have high expression of virally-encoded miRNAs that target BIM (Marquitz et al., Virology 2012), one of the putative targets identified by the authors. EBV is believed to 'push' infected cells through a GS-like reaction (Thorley-Lawson et al., Curr Top Microbiol Immunol. 2015).

The point raised by the reviewer is certainly very interesting and has been included in the manuscript discussion (page 10, lines 13-17), as follows: “*Interestingly, the B lymphotropic Epstein-Barr (EBV) virus has been shown to promote the migration of infected naïve B lymphocyte to the GC (Thorley-Lawson, 2015). This correlates with high expression of virally encoded miRNAs that target several mRNAs, including PTEN that undergoes dramatic decreases at the protein level upon infection (Marquitz et al, 2012)*”.

Referee #2:

The study described in this manuscript examines transfer of miRNAs via EVs from T-cells to B-cells at the immune synapse, with the conclusion that T-cell EVs shuttle functional miRNAs to B-cells. Unfortunately, the data do not support the conclusions, the study is methodologically weak, and field guidelines are not followed. As for several previous studies in this genre, a close examination of the data suggests that serum contaminants or cellular fluctuations are a more likely explanation of the observations than selective miRNA loading and transfer via EVs.

Thank you very much for the thorough review of our manuscript. We have answered each of your points below and have added new experimental data that certainly improve the quality of the revised manuscript.

Major points:

- 1) It is unclear if the mouse model actually "worked" or could "work" since so many mature miRNAs persisted after the Dicer knockout. Mature miRNAs are very stable in some cells, lasting for up to several weeks. Also, the data support an apparent increase in some miRNAs even though there is little indication that EV transfer or even transfer of contaminants from cell culture is a major source of miRNAs. A careful study of miRNA persistence in this model would be a prerequisite to interpretation of the data.

We do agree with the reviewer that CD19-Cre^{Ki/+} DICER^{fl/fl} (DICER-KO) cells can retain low levels of miRNAs, as previously described (Belver et al, 2010; Harfe et al, 2005) and also indicated by the RNA sequencing data in this study. Although we acknowledge that this could represent a limitation of the model used, small RNA sequencing revealed that the increase of the three miRNAs was found when comparing the -OVA and +OVA

conditions, while low levels of these miRNAs were detected in B cells in the absence of a mature OVA-specific IS (new Expanded View Fig. EV2C).

Moreover, we have quantified by qPCR the relative levels of the specific transferred miRNAs under study in WT and DICER-KO B cells before and after stimulation with LPS and IL4 (new Expanded View Fig. EV2D). This analysis showed that isolated DICER-KO B cells exhibited very low levels of mmu-miR-20a-5p, mmu-miR-25-3p and mmu-miR-155-3p (EV2D), which were not increased upon LPS+IL4 stimulation. Also, mmu-miR-20a-5p and mmu-miR-155-3p, but not mmu-miR-25-3p, levels increased upon B cell stimulation with LPS and IL4 only in WT B lymphocytes (mmu-miR-20a-5p, fold-increase: 2,19; mmu-miR-155-3p, fold-increase: 8,16; mmu-miR-25-3p, fold-increase: 0,83). This indicates that the upregulation of the identified miRNAs after an OVA-specific IS could not be attributed just to B cell stimulation, and it clearly does not account for miRNA increase in DICER-deficient B cells. Our LPS and IL4 induction results are in agreement with previously published data, showing that the three identified miRNAs were 2-4 fold more expressed in GC B cells compared to total mature B lymphocytes (Kuchen et al, 2010).

Additionally, we have included the data showing similar levels of DICER mRNA in our experimental conditions, excluding an eventual overgrowth of DICER non-depleted cells in OVA vs non-OVA cocultures (new Expanded View Fig. EV2B).

We have added all this additional supplementary information in the figures and modified the text (page 6, lines 7-20) and figure legends accordingly, as follows: *“We focused on the 3 miRNAs conserved in humans: mmu-miR-20a-5p, mmu-miR-25-3p, and mmu-miR-155-3p (Fig EV2A). Although DICER-KO B cells can retain low levels of miRNAs, as previously described (Belver et al, 2010; Harfe et al, 2005), small RNA sequencing revealed that the increased levels of mmu-miR-20a-5p, mmu-miR-25-3p, and mmu-miR-155-3p were only found after IS formation (Fig EV2C). In addition, we observed no changes in residual Dicer expression in DICER KO B cells upon OVA exposure (Figure EV2B). Moreover, in the absence of T cell contact, B cells from DICER KO (CD19Cre^{Ki/+} Dicer^{fl/fl}) mice showed very low levels of mmu-miR-20a-5p, mmu-miR-25-3p and mmu-miR-155-3p (Fig EV2D) both before and after stimulation with LPS and IL4. Also, mmu-miR-20a-5p and mmu-miR-155-3p increased upon LPS+IL4 B cell stimulation in WT B lymphocytes but not in DICER KO B cells, while mmu-miR-25-3p was downregulated after stimulation, in agreement with previously published data (Kuchen et al, 2010). Thus, miRNA upregulation in DICER-deficient B cells cannot be attributed to B cell stimulation, and instead is triggered by the OVA-specific IS.”*

Concerning the possibility of cell culture miRNA contaminants, the same batch of EXO-free medium was used and characterized before experimentation by Nanosight. We have added the following sentence to the material and method section (page 11, lines 15-16): *“The same batch of ultracentrifuged EV-depleted serum was used for all experiments to minimize variability and checked by NanoSight for EV depletion.”* It is also important to point out that the cell culture medium was the same in all the experimental conditions and that we statistically compared the -OVA/+OVA conditions.

Moreover, both culture medium and supernatants from CD4+ T cells derived from OT-II mice activated for 16 h in the presence of OVA peptide (experimental condition in our model) were ultracentrifuged, followed by PBS wash. Ultracentrifuged small EV pellets were analyzed by qPCR showing that the three identified microRNAs under study could not be detected in the culture medium alone (new Figure EV3F). See further details in point 3 below.

- 2) Use of the term "exosome" in the title and throughout the manuscript is unsupported, and it seems that the involvement of EVs is more assumed than supported. It should be EVs or small EVs, but even this is not well supported without more data. In addition to the perplexing and total lack of EV characterization, no specific evidence for exosome (vs other EV) involvement is shown. The authors are encouraged to consult with Lotvall et al, J Extracell Vesicles, 2014 and other recommendations of ISEV to improve design, nomenclature and reporting. Rab27a is now thought to be involved in EV biogenesis more broadly than just inside the endosome. Similarly, manumycin A and other purported exosome inhibitors have multiple modes of action and may work as well at the plasma membrane as in the endosomal system.

We completely agree with the reviewer that we did not include the data concerning the characterization of isolated T cell-derived extracellular vesicles in the manuscript, as a means to try to fit in a report format. We apologize for this and have now included the characterization of vesicles derived from OT-II CD4⁺ T cells (see point 3 below).

Also we acknowledge the broader effects of Rab27a and Manumycin that are not restricted to the endosomal system blockade and have changed the term "exosome" for "EVs or small EVs" throughout the manuscript.

3) The conclusions about specific EV loading into T-cell EVs are erroneous and not supported by the methods or the data. In fact, they are contradicted by the data. The EV separation method was ultracentrifugation, Although UC is still widely used and obviously has a continuing importance in the field, it does not efficiently purify EVs from other miRNA carriers, especially when no wash steps are performed, as here. The cells were cultured in medium supplemented with what is described as "EXO-free" serum but in reality this serum was not free of exosomes or other EVs. The serum was ultracentrifuged overnight, but this procedure is unable to remove all EVs. Additionally, no mention is made of diluting the serum appropriately before dilution or confirming removal of EVs, casting more doubt on the suitability of this serum. Please consult Driedonks et al, JEV, 2019 on the difficulties of removing EVs from serum.

Even if the serum were completely EV-depleted, most miRNA in serum is not in EVs. Centrifugation methods used in this paper will give pellets that contain both EV and non-EV miRNAs.

If one examines the TPM table for T-cells versus EVs, contamination of the EV fraction with serum miRNAs is clear, see Krichevsky's paper on serum RNA artifacts. For example, red blood cell-specific miR-451 and miR-144 show up only in the EVs. Stroma-specific miR-143 and miR-145 are only in the EVs or, for miR-143-3p, at hundreds of reads in the EVs vs none or single digits in the cells. Similarly, miR-486 is highly abundant in the EVs but not in the cells. This evidence of contamination makes it impossible to credit the claim that miRNAs are specifically loaded into EVs, since one cannot know which EV miRNAs are contaminants and which are from the cells. (Harfe et al, 2005)

We have now further characterized T-cell derived EVs (new Expanded Fig. EV3). For this purpose, we have set up OT-II CD4⁺ T cell cultures as obtained in our IS co-culture model (see material and methods section). Supernatant from cells in culture was harvested and analyzed by two different EV isolation methods, ultracentrifugation, followed by PBS wash and size-exclusion chromatography (Fig. EV3B, see material and methods section for details).

Firstly, size-exclusion chromatography analysis of T cell culture supernatants as described in (Suarez et al, 2017), showed that all three microRNAs identified in the current study, mmu-miR-20a-5p, mmu-miR-25-3p and mmu-miR-155-3p, were mainly detected by qPCR in SEC Fraction 2, expressing CD63 and CD81 EV markers, and almost undetectable in the protein-enriched fractions (SEC Fraction 3), see Fig EV3C-E.

Furthermore, we also analyzed both CD4⁺T cell supernatants and EV-depleted supplemented culture medium by ultracentrifugation followed by PBS wash and performed qPCR

on the pellets obtained. As shown in Fig EV3F, all three microRNAs were present in the cell-culture supernatants while nearly undetectable in the ultracentrifuged culture medium alone. However, the red-blood cell miR-451a included as a control, could be detected both in ultracentrifuged medium and cell cultures. Altogether, we have shown that mmu-miR-20a-5p, mmu-miR-25-3p and mmu-miR-155-3p are present in EVs isolated from OT-II T cells, enriched in CD63 and CD81 markers.

We have added Expanded View EV3 and the following text (page 6, lines 21-27): *“Importantly, these miRNAs are expressed in activated T cells from wild-type C57BL/6 mice and in their secreted small EVs (Fig 2C and D, Appendix Table S2). Mmu-miR-20a-5p, mmu-miR-25-3p and mmu-miR-155-3p could also be detected in small EVs derived from OT-II CD4+ T cell cultures, isolated from CD63 and C81-expressing size-exclusion chromatography fractions and ultracentrifuged supernatants but not in culture medium alone, excluding possible serum contaminations in these samples (Fig EV3).”*

Also the following paragraph was added to the material and method section (page 14, lines 18-24): *“For EV characterization, splenocytes and lymph nodes were harvested from OT-II mice as described above. Supernatants from 72h CD4+ T cell cultures were processed by ultracentrifugation as previously described, followed by PBS wash, and size-exclusion chromatography, as described in (Suarez et al, 2017) for analysis of small EVs and microRNAs content.”*

4) This is related to the previous point, but I cannot help but come away with the impression that there was little basis for the selection of the chosen miRNAs. Ranking the miRNAs by differential abundance in B-cells across the conditions, there were numerous miRNAs for which a better case could be made. Some of the selected miRNAs are also indicative of fundamental problems with the analysis. To give some examples:

miR-1946a was detected in all B-cell samples, and at a respectable average of just under 100 TPM. This miRNA also underwent a relatively large fold change in co-cultured versus alone. However, it was detected in only one T-cell/"EV" pair, and there at less than 10 TPM. Also, lower in the EV sample than in the T-cells. Strangely, Figure 3D still shows this miRNA and even indicates variance when there is none (only one positive sample!).

miR-155-3p was detected at >900 TPM in the B-cells, but had negligible levels in the T-cells and was detected in only 2 of 3 T-cell EV samples.

miR-6399 has an average of 72 TPM in B-cells, but was detected at single-digit TPM (3) in only one T-cell sample and no T-cell EVs.

miR-143-3p, despite having hundreds of TPMs in the EV fraction from T-cells, went from 0 to 6 in the B-cells in one experiment, 6 to 0 in another, and 49 to 74 in the third. There was no clear RNA transfer.

miR-486-5p, despite having nearly 900 TPM in the EV fraction from T-cells, increased just as much in the "alone" B-cells as in the co-culture condition. If anything, this suggests that some serum contaminants can skew the results of the experiments, and that these contaminants have more of an effect than the actual T-cell EVs.

Many miRNAs undergo a decline in OVA+ co-culture but not without co-culture. These include miR-15a, miR-191, and miR-10b. These are not mentioned or explained.

These examples suggest that much more is happening in the B-cells under these experimental conditions than EV transfer alone, and that miRNAs can undergo an apparent "increase" even in these Dicer-negative cells. Also that EV transfer from T-cells probably does not explain much if any of the observed miRNA fluctuations.

We are now aware that a description of the methodology used for small RNASeq data analysis was not included in the "Materials and Methods" section and we apologize for the missing information, and acknowledge that this has led to a certain level of confusion regarding how candidate miRNAs were chosen.

A subsection entitled "RNA extraction, real-time-PCR, small RNA next generation sequencing (NGS) and data analysis" contains now a description of how small RNASeq data was

processed. What follows is a version of such description, slightly expanded to provide additional details of how unbiased differential expression tests were performed.

Small RNAseq data was analyzed by the Bioinformatics Unit at CNIC. Briefly, sequencing reads were processed with a pipeline that used FastQC, to assess read quality, and Cutadapt to trim sequencing reads, eliminating Illumina adaptor remains, and to rule out reads that were shorter than 20 bp or had not been trimmed at all. Resulting reads were aligned against a mouse sequence database consisting of the mature miRNA component of miRBase21, using parameters optimized for very short sequence alignments, with either bowtie or BWA. Expression was then quantified with RSEM, to obtain matrices consisting in either raw counts or TPM (transcripts per million). To deal with multimapper reads (those mapping on more than one sequence in the reference database), RSEM uses an expectation-maximization algorithm to calculate maximum likelihood abundance estimates. Henceforth, RSEM calculated counts should be considered in fact "expected raw counts", more than "raw counts".

Expected raw count matrices were then processed with a differential expression analysis pipeline that used Bioconductor package EdgeR for normalization and differential expression testing. The method chosen for normalization with EdgeR, TMM (weighted trimmed mean of M-values), calculates scaling factors for each sample after discarding those genes (or small RNAs) that are expressed at extremely high levels, to prevent underestimating the expression of the remaining genes. Scaling factors are applied by EdgeR to estimate what are known as "effective library sizes", which are then used in all downstream analyses, including the calculation of normalized counts per million (CPM). Differential expression tests were performed only on those small RNAs expressed at a minimal level of 1 CPM, in a number of samples equal to the number of replicates of the condition with less replicates. An essential step to identify significant differences in expression levels is to have good estimates of expression variance. This is problematic in most RNAseq experiments given the relatively small number of replicates for each condition. EdgeR, and other RNAseq analysis tools, overcome this problem by using an empirical Bayes approach that allows sharing dispersion information across genes, before fitting a negative binomial model and performing gene-wise statistical tests for a given contrast. When required, a blocking variable was used to define groups of samples that were expected to be similar. To correct for multiple testing, p-values were adjusted with the Benjamini and Hochberg method, and the threshold for changes in small RNA expression was set to adjusted p-value < 0.2 .

For the sake of clarity, we have replaced supplementary Tables 1 and 2 (Appendix), which originally described TPM values, with new tables that contain the output directly produced by the differential expression pipeline. Information is now restricted to those miRNAs that were detected as expressed in each RNAseq experiment (around 500), following the definition indicated in the previous paragraph. Fields include, for each miRNA, averaged normalized expression in each condition and each sample (expressed as CPM), log-fold change for the corresponding contrast, p-value and adjusted p-value. Tables are sorted by adjusted p-value and the three independent experiments are shown, indicating paired-wise analysis of -OVA and +OVA conditions and fold-change.

In supplementary Table 1, the top seven miRNAs are highlighted with yellow background. These are the seven miRNAs for which changes in log fold change between the -OVA and +OVA conditions had an adjusted p-value equal or lower than 0.2. These correspond to the six upregulated miRNAs described in the manuscript and a seventh which is downregulated.

In supplementary Table 2, 178 miRNA log fold change values are associated to adjusted p-values lower than 0.2.

From the differentially expressed miRNAs we continued the analysis only with those miRNAs that had human counterparts (excluding mmu-miR-1946a-5p, mmu-miR-6399 and mmu-miR-147-3p) and could be detected in T-cell derived exosomes (excluding mmu-miR-1946a-5p detected in only one T-cell/EV pair as pointed out by the reviewer and mmu-miR-6399). Taking this into account we continued the study with mmu-miR-20a-5p, mmu-miR-25-3p and mmu-miR-155-3p.

To further confirm the expression of the identified microRNAs in T-cell exosomes, size-exclusion chromatography and ultracentrifugation followed by PBS wash were performed on OT-II CD4+ T cell culture supernatants confirming the presence of these specific microRNAs in small EVs, cofractionating with the exosomal markers CD63 and CD81 (see point 3 above).

Concerning the potential contribution of serum contaminants, please refer to the previous points above. Also it is important to point out that these potential serum contaminants would be present in all samples and thus they should not affect the +OVA vs –OVA comparison.

5) Sequencing data should be deposited with a public database. While the data can be embargoed until publication, a working link should be provided for reviewers. The spreadsheets are helpful, but do not replace the need to deposit raw data.

The data discussed in this publication have been deposited in NCBI's Gene Expression Omnibus and are accessible through GEO Series following accession numbers:

microRNA expression changes in DICER-KO B cells after OVA-specific immune synapse formation

- Accession number: GSE140981
- Password: ivgvamecdnkbxal

microRNA repertoire of wild-type C57BL/6 mice T cell blasts and their secreted small extracellular vesicles (EVs)

- Accession number: GSE140982
- Password: arahiikoztodjmf

All the data were included in the super-series:

Extracellular vesicle-microRNA transfer controls germinal center reaction and antibody production

- Accession number: GSE141045
- Password: gbezoqasppcxhur

Minor points: We agree with all the minor points raised by the reviewer and have changed the figures and text accordingly.

While many readers will be familiar with immunology, and will in any case learn more as they read deeper into the manuscript, it would be helpful for the broader readership to define the cells and interactions more explicitly in the abstract and introduction. Instead of "T-B" and "B-T", start with "contacts between B lymphocytes and T lymphocytes" and also keep the order consistent.

The figures require more attention. For a few examples:

Bar graphs are less informative than plotting the actual data points.

Red/Green (Fig 2 A, C) should be replaced by colors that are distinguishable by the colorblind.

Figure 2B is misleading, since fold change of 1=no change. So the axis should be at 1 not 0. Better would be to show % increase. Also, variance is not shown.

Minor edits: We agree with all the minor edits and have changed the text accordingly.

Intro

"while maintaining homeostasis": what homeostasis?

bio-molecules: no hyphen needed

Results

"To analyze the effects on B cell activation and class-switch recombination (CSR)

in vitro, we used alternate B cell stimuli": I believe you mean "alternative" not "alternate"

"IS-dependent transfer of a specific set of microRNAs from the T to the B lymphocyte" section: the first sentence is repeated from the intro and should be removed or shortened.

Methods

Nanosight is "NanoSight" and Laemli is "Laemli"

Referee #3:

Fernandez-Messina et al. investigate the requirement of T cell-derived exosomal microRNAs through the immunological synapse in the activation of B cells in vitro and in germinal center (GC) development in vivo. The authors report that antigen-dependent B cell development is controlled by this mechanism, and identify a set of six microRNAs as the underlying biological mechanism for the activation. The in vivo role of the interaction is demonstrated by exosome-deficient T cells in immunized mice undergoing the GC reaction.

There are several experimental issues that affect the interpretation of the results. While conceptually interesting, as presented, the work has preliminary character.

We thank the reviewer for his/her important comments and suggestions. We have now addressed all the questions raised, and replied to his/her queries in the point-by-point reply below.

- 1) CD19-Cre mice were used to delete Dicer in Dicer-floxed B cells. CD19-Cre mice delete the floxed allele only in about 90% of B cells, which has the implication that a sizable fraction of cells are not deleted, which affects single cell assays as the ones shown in Fig. 1A. It is necessary to demonstrate that the B cells forming immunological synapses with the T cells are indeed Dicer-deficient and do not represent contaminating non-deleted B cells.

For the experiments with cell populations, also here the incomplete deletion issue needs to be addressed, although the results would suggest that it is unlikely that the difference is due to a preferential reactivity or outgrow of the 10% non-deleted cells. A response to this critical issue would be appropriate, and it should be addressed in writing to pre-empt any thoughts from a critical reader.

We do agree with the reviewer that DICER-floxed B cells are only depleted in about a 90% of the cells and that non-depleted B cells may be influencing the observed results. To monitor for an eventual overgrowth of post-synaptic non-depleted B cells we have performed quantitative PCR to determine DICER mRNA levels in the presence or absence of OVA peptide and therefore of a mature IS. As shown in the supplementary Expanded Figure EV2, the levels of DICER mRNA before and after IS are not affected. We have included this figure and the next sentence was added to the manuscript (page 6, lines 11-21): *"We focused on the 3 miRNAs conserved in humans: mmu-miR-20a-5p, mmu-miR-25-3p, and mmu-miR-155-3p (Fig EV2A). Although DICER-KO B cells can retain low levels of miRNAs, as previously described (Belver et al, 2010; Harfe et al, 2005), small RNA sequencing revealed that the increased levels of mmu-miR-20a-5p, mmu-miR-25-3p, and mmu-miR-155-3p were only found after IS formation (Fig EV2C). In addition, we observed no changes in residual Dicer expression in DICER KO B cells upon OVA exposure (Figure EV2B). Moreover, in the absence of T cell contact, B cells from DICER KO (CD19CreKi/+ Dicerfl/fl) mice showed very low levels of mmu-miR-20a-5p, mmu-miR-25-3p and mmu-miR-155-3p (Fig EV2D) both before and after stimulation with LPS and IL4. Also, mmu-miR-20a-5p and mmu-miR-155-3p increased upon LPS+IL4 B cell stimulation in WT B lymphocytes but not in DICER KO B cells, while mmu-miR-25-3p was downregulated after stimulation, in agreement with previously published data (Kuchen et al, 2010). Thus, miRNA upregulation in DICER-deficient B cells cannot be attributed to B cell stimulation, and instead is triggered by the OVA-specific IS."*

- 2) Fig. 3F: GL7 staining alone cannot be used as a measure for GC B cells. It need be used in combination with CD38 or CD95, or PNA. Only double stained cells, such as GL7+CD38-low or GL7+CD95+ or PNA+ identifies GC B cells. Since the enumeration of GC B cells is

the end-point of the analysis, a rigorous analysis of GC B cells numbers and percentages by flow, and GC structure by immunohistochemistry of tissue sections, would be appropriate to convince that GC development is affected.

We agree with the reviewer that GL7 staining alone is not enough to define GC B cells and have re-analyzed the data including, as suggested by the reviewer, CD95 (Fas) antibody to discriminate GL7⁺ CD95⁺ germinal center B cells. We have changed the figures (new Figure 3F, left: Germinal Centre and right: quantification and statistical analysis), text and figure legends accordingly. Unfortunately, immunohistochemistry studies were not performed, given that the use of irradiated Rag deficient mice has an impact on the follicle architecture.

The following paragraph (page 9, lines 12-14) was re-written: “*Furthermore, GC Fas⁺GL7⁺ B cells of mice adoptively transferred with Rab27-KO CD4⁺ T cell had an altered dark-zone: light-zone ratio (Fig 3G), consistent with dysfunctional GC dynamics.*”

- 3) Fig. 3G: The staining does not identify dark zone and light zone B cells correctly. Light zone B cells are CD86+CXCR4-minus, and dark zone cells CXCR4+CD86-minus. Therefore, also the summary plots are incorrect, and no conclusions can be drawn from this experiment. For how the staining should look like, please refer to the original Victoria Cell and Blood papers where this staining has first been described.

We have re-analyzed the data, gating on double positive GL7⁺ Fas⁺ germinal center B cells after *in vivo* immunization. Plots and statistical analysis were included in the new Figure 3G, and text and figure legends modified accordingly.

- 4) Fig. 3H: Measuring total serum IgM and IgG is not the adequate approach to measure GC output. For this, the immunization with, and specific detection of antibodies against, a model antigen would be required.

We agree with the reviewer that serum levels of IgM and IgG are not a direct readout for GC reaction. For this reason, in the text we will rather refer to the “antibody immune response” than to a GC response read out. The text has been modified as follows (page 9, lines 14-17): “*Finally, serum ELISA showed a significant reduction in the production of both soluble IgM (Fig 3H) and class-switched IgG (Fig 3I), revealing the importance of EV-dependent B and T cell communication for antibody responses in vivo.*”

REFERENCES:

Belver L, de Yebenes VG, Ramiro AR (2010) MicroRNAs prevent the generation of autoreactive antibodies. *Immunity* **33**: 713-722

Harfe BD, McManus MT, Mansfield JH, Hornstein E, Tabin CJ (2005) The RNaseIII enzyme Dicer is required for morphogenesis but not patterning of the vertebrate limb. *Proc Natl Acad Sci U S A* **102**: 10898-10903

Kuchen S, Resch W, Yamane A, Kuo N, Li Z, Chakraborty T, Wei L, Laurence A, Yasuda T, Peng S, Hu-Li J, Lu K, Dubois W, Kitamura Y, Charles N, Sun HW, Muljo S, Schwartzberg PL, Paul WE, O’Shea J, Rajewsky K, Casellas R (2010) Regulation of microRNA expression and abundance during lymphopoiesis. *Immunity* **32**: 828-839

Marquitz AR, Mathur A, Shair KH, Raab-Traub N (2012) Infection of Epstein-Barr virus in a gastric carcinoma cell line induces anchorage independence and global changes in gene expression. *Proc Natl Acad Sci U S A* **109**: 9593-9598

Mittelbrunn M, Gutierrez-Vazquez C, Villarroja-Beltri C, Gonzalez S, Sanchez-Cabo F, Gonzalez MA, Bernad A, Sanchez-Madrid F (2011) Unidirectional transfer of microRNA-loaded exosomes from T cells to antigen-presenting cells. *Nat Commun* **2**: 282

Suarez H, Gamez-Valero A, Reyes R, Lopez-Martin S, Rodriguez MJ, Carrascosa JL, Cabanas C, Borrás FE, Yanez-Mo M (2017) A bead-assisted flow cytometry method for the semi-quantitative analysis of Extracellular Vesicles. *Sci Rep* **7**: 11271

Thorley-Lawson DA (2015) EBV Persistence--Introducing the Virus. *Curr Top Microbiol Immunol* **390**: 151-209

2nd Editorial Decision

17 January 2020

Thank you for the re-submission of your revised manuscript to our editorial offices. We have now received the reports from the three referees that were asked to re-evaluate your study, you will find below. As you will see, the referees now support the publication of your study in EMBO reports. Nevertheless, there are some final concerns and suggestion to be addressed. Please address the comment by referee #1 by additions to the text, or by discussing the points mentioned. Please also address the comments of referee #3, in particular his/her points 3-5. It might be advisable to add further data (replicates) to strengthen the data. Please also provide a point-by-point-response to the remaining concerns of both referees).

Further, I have these editorial requests:

- I suggest this modified title (any more compact title suggestion would be welcome):
Transfer of extracellular vesicle-microRNA controls germinal center reaction and antibody production
- Please reduce the number of keywords to five.
- Please ensure that primary datasets produced in this study (e.g. RNA-seq, ChIP-seq and array data) are deposited in an appropriate public database. See:
<http://embor.embopress.org/authorguide#datadeposition>

The accession numbers and database should be listed in a formal "Data Availability " section (placed after Materials & Methods) that follows the model below. Please note that the Data Availability Section is restricted to new primary data that are part of this study.

Data availability

- In the author contributions, it seems Sandra Tenreiro and Miguel Seabra are missing. Please check and provide their contributions.

- Please format the references according to our journal style. See:
<http://www.embopress.org/page/journal/14693178/authorguide#referencesformat>

- Appendix Tables S1 and S2 are too long to be included in the Appendix. They need to be datasets (data files that will be linked to the article). Please upload these files as Dataset files, named Dataset

EV1 and Dataset EV2. Please add the legends for these datasets as a new TAB to the respective excel file (as first TAB). Please also update the callouts for these files in the manuscript text.

- Appendix Table S3 is fine, but needs to be supplied as part of a single pdf labeled Appendix. The Appendix should have page numbers and needs to include a table of content on the first page (with page numbers) and legends for all content. Please follow the nomenclature Appendix Figure Sx, Appendix Table Sx etc. throughout the text, and also label the figures and tables according to this nomenclature. I suggest that this table is then Appendix Table S1.

- As it does not contain main data, I suggest to move the present Table EV1 to the Appendix. Please put this table together with a legend into the Appendix file, and name it according the nomenclature Appendix Table SX and include it also in the TOC of the Appendix file. I suggest that this table is then Appendix Table S2. Finally, please update the callouts for this table in the manuscript file. Or, add a call out for this table.

- In the methods section, in the paragraph 'In silico target analysis', a Table 3 is mentioned. Please use the correct call-out here (see above).

- The legend of Fig. 2 lists panels I and J twice. Please check and combine.

- Regarding data quantification and statistics, can you please assure that, where applicable, the number "n" for how many independent experiments (biological vs. technical replicates) were performed, the bars and error bars (e.g. SEM, SD) and the test used to calculate p-values is specified in the respective figure legends. Please provide statistical testing where applicable, and also add a paragraph detailing this to the methods section. See:
<http://www.embopress.org/page/journal/14693178/authorguide#statisticalanalysis>

- Finally, please find attached a word file of the manuscript text (provided by our publisher) with changes we ask you to include in your final manuscript text, and some queries, we ask you to address. Please provide your final manuscript file with track changes, in order that we can see the modifications done.

In addition I would need from you:

- a short, two-sentence summary of the manuscript
- two to three bullet points highlighting the key findings of your study
- a schematic summary figure (in jpeg or tiff format with the exact width of 550 pixels and a height of not more than 400 pixels) that can be used as a visual synopsis on our website.

REFeree REPORTS

Referee #1:

The authors adequately responded to my questions. As I suspected the problems raised on possible serum contamination issues, though critical to exclude, were adequately mitigated. Indeed, the fact that IS formation supports transport of specific miRNAs is very relevant to the field and in my view convincingly shown.

I do have one final comment. Recent advances strongly suggest that miRNA function is much more complex than previously thought (Gebert et al., Nat Rev Mol Cell Biol 2019). It is possible, if not likely, that also other (mi)RNAs transported via the T cell exosomes have a role in recipient B cell biology. Also it should be mentioned that Nebb-next Illumina sequencing protocols have a strong (ligation+PCR) bias for particular (mi)RNAs casting doubt on EV-enrichment scores when based on small RNAseq data alone.

Referee #2:

Overall, the quality of the manuscript has been improved on revision, and while I have several remaining concerns, I do not think they are so important as to prevent a decision.

Referee #3:

Point 1: The critique was focused on the IF shown in Fig. 1. It cannot be said with certainty that the B cell shown is Dicer deleted. While this was not directly addressed, the additional experiments shown in Figure EV2 at least rule out an outgrowth of Dicer-proficient (i.e. non-deleted) cells, and the summary graph in Fig. 1B makes it unlikely that all of the cells are Dicer-proficient.

Point #2: Has been adequately addressed.

Point #3: The requested analysis has been performed. The outcome of this was a dramatic change in the percentage of GC among WT (from 12% to 5%) and Rab27KO (from 8% to 4%), implying that the observed difference, though significant, is rather small.

Point #4: The staining is improved, although there is still no CD86+ population that is CXCR4-negative. It is a bit worrisome to get from the figure shown in the previous submission to that one with the same data. To me, the results appear to be random.

Point #5: The authors decided to change their claims in the text rather than provide a more specific detection of antibodies against a model antigen, so the results that show a minor difference with questionable biological significance have to be taken with caution.

Overall, I am not convinced from the data I see in the manuscript that extracellular vesicle-microRNA transfer has a role in the germinal center reaction, as the paper claims. But since the authors addressed my criticism, and since opinions may vary, perhaps the work could be presented in its present form to the community to let them judge.

2nd Revision - authors' response

21 January 2020

Referee #1:

The authors adequately responded to my questions. As I suspected the problems raised on possible serum contamination issues, though critical to exclude, were adequately mitigated. Indeed, the fact that IS formation supports transport of specific miRNAs is very relevant to the field and in my view convincingly shown. I do have one final comment. Recent advances strongly suggest that miRNA function is much more complex than previously thought (Gebert et al., Nat Rev Mol Cell Biol 2019). It is possible, if not likely, that also other (mi)RNAs transported via the T cell exosomes have a role in recipient B cell biology. Also it should be mentioned that Nebb-next Illumina sequencing protocols have a strong (ligation+PCR) bias for particular (mi)RNAs casting doubt on EV-enrichment scores when based on small RNAseq data alone.

We thank the referee for his/her thorough review and comments that have certainly improved the quality of the manuscript.

We agree with the reviewer that miRNA function is complex and we can not rule out the possibility that other miRNAs contained in T cell exosomes can have a role in the post-synaptic B lymphocyte. To emphasize this idea, the following sentence has been added to the manuscript (Page 6, Lines 9-11): "It is worthwhile mentioning that, although we focused on the upregulated microRNAs identified in our study, it is likely that other T-cell EV-microRNAs may also have a role in recipient B lymphocytes."

Also, we acknowledge that Neb-next Illumina sequencing protocols can have a bias for particular (mi)RNAs, thus EV-enrichment ratios based on small RNA seq may be taken carefully. According

to the reviewer's comment, the following sentence has been included in the revised version of the manuscript (Page 6, Line 30-31): "The bias of small RNA sequencing protocols for particular microRNAs should be taking into account when analyzing the EV-enrichment scores (Fig EV3A)."

Referee #2:

Overall, the quality of the manuscript has been improved on revision, and while I have several remaining concerns, I do not think they are so important as to prevent a decision.

We very much thank the reviewer for his/her in depth review that has certainly contributed to improve the manuscript upon revision.

Referee #3:

Point 1: The critique was focused on the IF shown in Fig. 1. It cannot be said with certainty that the B cell shown is Dicer deleted. While this was not directly addressed, the additional experiments shown in Figure EV2 at least rule out an outgrowth of Dicer-proficient (i.e. non-deleted) cells, and the summary graph in Fig. 1B makes it unlikely that all of the cells are Dicer-proficient.

We thank the reviewer for his/her comment and acknowledge the importance of ruling out the possibility of an outgrowth of DICER-proficient cells.

Point 2: Has been adequately addressed.

We thank the reviewer for his/her comment.

Point 3: The requested analysis has been performed. The outcome of this was a dramatic change in the percentage of GC among WT (from 12% to 5%) and Rab27KO (from 8% to 4%), implying that the observed difference, though significant, is rather small.

We agree with the reviewer that the percentage of GC has changed after re-analysis, as a result of analyzing only double positive GL7+Fas+ instead of single positive GL7+, as suggested by the reviewer, thus allowing a more accurate definition of GC B cells.

Point 4: The staining is improved, although there is still no CD86+ population that is CXCR4-negative. It is a bit worrisome to get from the figure shown in the previous submission to that one with the same data. To me, the results appear to be random.

We agree with the reviewer that CXCR4 staining generally allows better resolution than CD86 expression for LZ/DZ definition. However, we observe a statistically significant increase in CD86 expression (Mean fluorescence intensity) in LZ GC B lymphocytes in control animals (Figure R1).

Figure R1. CD86 and CXCR4 expression in dark-zone and light zone GC B cells. Analysis of CXCR4 and CD86 expression (flow cytometry Mean Fluorescence Intensity, MFI) on B cells for Dark zone (DZ) and Light zone (LZ) analysis (gated on GC GL7+ Fas+ B cells). Significance was assessed by paired Student t test comparing the DZ and LZ populations; * $P < 0.05$, ** $P < 0.01$, *** $P < 0.001$.

Point 5: The authors decided to change their claims in the text rather than provide a more specific detection of antibodies against a model antigen, so the results that show a minor difference with questionable biological significance have to be taken with caution Overall, I am not convinced from the data I see in the manuscript that extracellular vesicle-microRNA transfer has a role in the germinal center reaction, as the paper claims. But since the authors addressed my criticism, and since opinions may vary, perhaps the work could be presented in its present form to the community to let them judge.

We agree with the referee that a specific detection of antibodies against a model antigen will strengthen the data and provide additional information and acknowledge that further studies will be needed to dissect the role of EV-microRNA transfer in post-synaptic B cells.

Accepted

24 January 2020

I am very pleased to accept your manuscript for publication in the next available issue of EMBO reports. Thank you for your contribution to our journal.

Corresponding Author Name: Francisco Sánchez-Madrid

Manuscript Number: #EMBOR-2019-48925V2